

# Surface drifters in the German Bight: Model validation considering windage and Stokes drift

Ulrich Callies[1], Nikolaus Groll[1], Jochen Horstmann[1], Hartmut Kapitza[1], Holger Klein[2],
Silvia Maßmann[2], and Fabian Schwichtenberg[2]

[1]Helmholtz-Zentrum Geesthacht, Max-Planck-Str. 1, 21502 Geesthacht, Germany
[2]Federal Maritime and Hydrographic Agency (BSH), Bernhard-Nocht-Str. 78, 20359 Hamburg, Germany
*Correspondence to:* U. Callies (ulrich.callies@hzg.de)

**Abstract.** Six surface drifters (drogued at about 1 m depth) deployed in the inner German Bight (North Sea) were tracked for between 14 and 54 days. Corresponding simulations were conducted offline based on surface currents from two independent models (BSHcmod and TRIM). Inclusion of a direct wind drag (0.6 % of 10 m wind) was needed for successful simulations based on BSHcmod currents archived for a 5 m depth surface layer. Adding 50 % of surface Stokes drift simulated with the third generation wave model WAM was tested as an alternative approach. Results resembled each other during most of the time. Successful simulations based on TRIM surface currents (1 m depth) suggest that both approaches were mainly needed to compensate insufficient vertical resolution of hydrodynamic currents.

The study suggests that main sources of simulation errors were inaccurate Eulerian currents and lacking representation of sub-grid scale processes. Substantial model errors often occurred under low wind conditions. A lower limit of predictability (about 3-5 km per day) was estimated from two drifters that were initially spaced 20 km apart but converged quickly and diverged again after having stayed at a distance of 2 km and less for about 10 days. In most cases, errors in simulated 25 h drifter displacements were of similar order of magnitude.

## 1 Introduction

Lagrangian particle tracking is a natural choice when origins or destinations of drifting objects (or water bodies) need to be known. Such methods have been developed for a wide range of applications (see Mariano et al., 2002). Examples from oceanography are simulations of physical dispersion (Schönfeld, 1995; Sentchev and Korotenko, 2005), possibly augmented by specific source and sink terms (e.g. Puls et al., 1997). In ecosystem modelling, Lagrangian transport models have been employed to better understand the process of non-indigenous species invading an ecosystem (Brandt et al., 2008), the risk of toxic algae blooms (Havens et al., 2010) or larval transport and connectivity being crucial to spatial fishery management (e.g. Nicolle et al., 2013; Robins et al., 2013). Lagrangian transport simulations provide also a basis for more comprehensive individual-based models of fish recruitment (e.g. Daewel et al., 2015), for instance.



Obviously, the quality of Lagrangian drift simulations has a particularly high practical relevance in the context of emergency operations like search and rescue (Breivik et al., 2013) or organizing efficient combating of oil spills (Broström et al., 2011; Maßmann et al., 2014). Modelling of surface drifter trajectories is particularly challenging as many input factors needed are poorly known. Often drift properties of the search object can only be estimated (Breivik et al., 2013). The present study refers to a drifter experiment conducted in the inner German Bight (North Sea) during May-July 2015. Corresponding offline drift simulations based on archived currents from two different models were undertaken to assess the degree of uncertainty that must reasonably be expected in this region.

The surface drifters deployed are ideal in the sense that their exposure to a direct aerodynamic force from wind (leeway or windage (Breivik and Allen, 2008)) seems negligible. However, already Eulerian surface currents used can be a major source of uncertainty. The circulation model BSHcmod, this study mainly focusses on, is run operationally. In cases of necessity drift simulations will be based on an re-gridded archived version of model predictions with near surface currents representative for a 5 m depth surface layer. Therefore even for an ideal surface drifter, introducing a direct wind drag can be helpful as a means of compensating insufficient vertical resolution of marine currents. The second hydrodynamic model employed in this study, TRIM, was set up with a 1 m surface layer. Comparing drift simulations based on outputs from the two different models helps assess uncertainties possibly related to the vertical resolution of near surface currents.

More complex impacts of winds on surface currents may be mediated via waves (Perrie et al., 2003; Ardhuin et al., 2009). Röhrs et al. (2012) found evidence that predictability of drift trajectories can be improved by the inclusion of numerical wave modelling. On the other hand, Stokes drift and other wave effects are often neglected in operational systems. According to Breivik and Allen (2008) the main reason for this is that wave processes are already taken into account by empirically tuned windage coefficients that summarize the deflection of an object's trajectory induced by combined impacts of both winds and waves. The situation can differ in nearshore regions, where wave refraction directs wave induced transports towards the coast (Sobey and Barker, 1997).

A key objective of this study is checking whether inclusion of Stokes drift calculated with the state-of-the-art wave model WAM improves simulations. Although provided with consistent atmospheric forcing, the wave model is run in a stand alone mode rather than fully coupled as in Staneva et al. (2016), for instance. Assessing the necessity to distinguish between effects of a direct wind drag and Stokes drift is essential to avoid over-parametrization.

Horizontal grid resolutions of the two hydrodynamic data sets (900 m in BSHcmod and 1.6 km in TRIM) allow for a proper representation of mesoscale eddies in the region of interest. However, simulations may miss relevant sub-mesoscale processes. According to Kjellsson and Döös (2012) the underestimation of eddy kinetic energy by a Eulerian flows is a common finding of many model validation studies. This deficiency could be fixed by a transition to an advection-diffusion equation, introducing an additional stochastic random walk term. In this context, specification of the proper eddy diffusivity as function of grid resolution poses a major problem. There are, however, also concerns regarding the simple theoretical concept. For the advection-diffusion approach being valid, a spectral gap should separate processes on the scale resolved from sub-grid scale processes. Such gap may often not exist (see De Dominicis et al., 2012, for instance).




Garraffo et al. (2001) compared the statistics of drifter observations in the North Atlantic with those of drift simulations based on Eulerian velocities from a model with about 6 km horizontal resolution. Without a stochastic model of sub-grid scale actions they found simulations to underestimate eddy energy. Simulated absolute dispersion being too low was also reported by Kjellsson and Döös (2012) evaluating drifters deployed in the Baltic Sea. Referring to global ocean data, Döös et al. (2011)

tuned random turbulent velocity in their drift model to achieve better agreement between relative dispersion of simulated trajectories and corresponding observations. However, they found this approach being too simple for a reasonable reproduction of Lagrangian properties.

More sophisticated analyses of the relative dispersion of pairs of particles try to distinguish the regimes of *local dispersion* driven by eddies comparable in size to the distance between two drifters and of *non-local dispersion* driven by eddies with

scales much larger than this distance (e.g. Koszalka et al., 2009). Beron-Vera and LaCasce (2016) conducted such an analysis for data from the Grand Lagrangian Deployment experiment (GLAD), in which more than 300 drifters were deployed in the Gulf of Mexico. Drifter launch positions spaced from 100 m to 15 km apart allowed to study sub-mesoscale dispersion characteristics in much detail. However, referring to experimental data in the southwestern Gulf of Mexico, Sansón et al. (2017) show that for large initial distances the probability density functions of pair separations get dependent on prevailing mesoscale

circulation patterns. This aspect seems particularly relevant for the present study, as variations of the residual current regime in the inner German Bight can very well be approximated in terms of only 2-3 degrees of freedom, depending on prevailing winds (Callies et al., 2017). Tidal currents dominate short term transports.

The data available for this study (six drifters, tracked between 9 and 54 days) do not provide a basis for studying features of oceanic turbulence. Therefore in the present model validation study, stochastic simulation of sub-grid scale processes will not

be considered. Ohlmann et al. (2012) provide an example that even an accurate reproduction of mean drifter pair separation does not necessarily imply good agreement between observations and corresponding simulations. According to Coelho et al. (2015), models used in the aforementioned GLAD experiment in the Gulf of Mexico had limited success capturing the observed drift patterns. Barron et al. (2007) provides a list of typical separation rates in different regions worldwide. For an experiment in the Ria de Vigo estuary in NW Spain, Huhn et al. (2012) reported simulation errors that were relatively small compared to

those typically found in the open ocean. This study tries to provide a realistic estimate of how reliable operational forecasts in the German Bight, another shelf sea region, can be expected to be. This includes gaining preliminary indications for regions where the deterministic part of a model needs improvement.

The paper is organized as follows: Section 2 documents how observations were taken (Sect. 2.1) and how corresponding model simulations were performed (Sect. 2.2). Section 2.3 describes two data sets characterizing residual currents variability

on a daily basis. Results (Sect. 3) are presented in two parts: Sect. 3.1 provides a synoptic description of all drifters deployed and places observations into the context of ambient atmospheric and marine conditions; Sect. 3.2 provides the analysis of how corresponding model simulations match observations. After simulations based on different model setups explored the range of possible effects, a more detailed evaluation of simulation errors is based on subdividing drift trajectories into segments of 25 h length. Results are discussed in Sect. 4, main conclusions provided in Sect. 5.



## 2 Material and Methods

### 2.1 Drifter observations

In May 2015 a total of nine drifters were deployed at different locations in the German Bight (North Sea) during the FS Heincke
cruise HE 445. The raw data are freely accessible at Carrasco and Horstmann (2017). Table 1 specifies each drifter's launch

position and launch time as well as its last position, the total lengths of its observed trajectory and the simple linear distance
between its initial and final location. Drifters #2, #3 and #4, travelling for only few days, were ignored for this study. All drifters
obtained their positions via the Global Positioning System (GPS) and communicated them to the lab via the global full ocean
coverage bidirectional satellite communication network Iridium. Three drifters could successfully be tracked for between 40
and 54 days.

Two different drifter types were utilized (cf. Table 1). The first drifter type, MD03i, from Albatros Marine Technologies
(Fig. 1(a)) is cylinder shaped with a diameter of 0.1 m and a length of 0.32 m. Only $\sim 0.08$ m of such drifter protrude from the
water surface when deployed (Fig. 1(b)). The second drifter type, ODi from the same manufacturer (Fig. 1(c)), has a spherical
shape with 0.2 m diameter, about half of it protruding from the water surface. To both drifters a drogue with 0.5 m length and
diameter (e.g. Fig. 1(a)) was attached 0.5 m below the sea surface. Due to this drogue and the small sail area exposed to winds

above the water surface, drifter movements are supposed to be representative for currents in a surface layer of about 1 m depth.

### 2.2 Drifter simulations with PELETS-2D

For drifter simulations we used the Lagrangian transport module PELETS-2D (Program for the Evaluation of Lagrangian
Ensemble Transport Simulations (Callies et al., 2011)) developed at Helmoltz-Zentrum Geesthacht (HZG). The PELETS al-
gorithm was designed for particle tracking on two-dimensional unstructured triangular grids. As both models underlying this

study use regular grids, the grid topology was pre-processed, splitting each rectangular grid cell into two triangles. Neither
the number of nodes nor the information content of underlying hydrodynamic fields is affected by this formal procedure. The
integration algorithm used is a simple Euler forward method. Particle velocities are updated (linear interpolation between two
neighbouring nodes) each time a particle leaves a cell of the triangular grid. If no edge is reached within the maximum time
step of 15 min, velocities are updated based on linear interpolation between three nodes.

The following equation is used for simulating drifter location $\boldsymbol{x}$ as function of time $t$:

$$\frac{d\boldsymbol{x}}{dt} = \boldsymbol{u}_E + \alpha \boldsymbol{u}_S + \beta \boldsymbol{u}_{10m} \tag{1}$$

Here $\boldsymbol{u}_E$ denotes the Eulerian marine surface currents calculated with either BSHcmod (Sect. 2.2.1) or TRIM (Sect. 2.2.2).
Components $\alpha \boldsymbol{u}_S$ and $\beta \boldsymbol{u}_{10m}$ represent additional contributions from Stokes drift and direct wind drag, respectively (Sect. 2.2.3).
Within this study, Stokes drift and wind drag will not be considered in combination but rather as alternative options. Therefore

at least one of the two weighting factors $\alpha$ and $\beta$ in Eq. (1) will always be set to zero.

Drift paths were calculated offline based on archived data. Sub-grid scale turbulence effects implemented in PELETS-2D in
terms of random movements were deactivated.

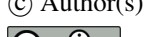



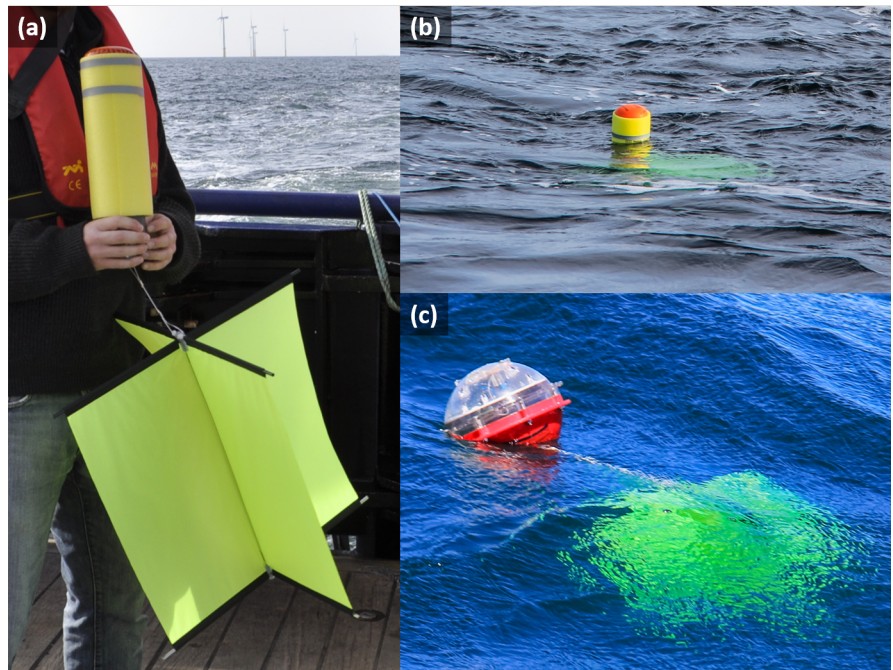

**Figure 1.** Drifter types MD03i (panels (a) and (b)) and ODi (panel (c)) used during the experiment. Both drifter types were photographed shortly after launch so that the drogue had not yet settled.

### 2.2.1 BSHcmod

BSHcmod is run operationally by the Federal Maritime and Hydrographic Agency (BSH). A description of the 3D model in geographic coordinates can be found in Dick and Kleine (2007). Spatial resolution in the German Bight is 900 m. Atmospheric forcing of BSHcmod is taken from the regional model COSMO-EU (Consortium for Small-Scale Modelling (Schulz and Schättler, 2014)). This operational atmospheric model of the German Meteorological Service (DWD) has a spatial resolution of 7 km.

Archived surface current data represent approximately the upper 5 m of the water column. Higher resolution output of the operational model BSHcmod (version 4) was re-gridded accordingly, conserving transport rates. Time resolution of archived data is 15 min. Although operationally BSHcmod is run in combination with its own Lagrangian transport module (Maßmann et al., 2014), for the present study this module was replaced by PELETS-2D which provides convenient interfaces to both BSHcmod and TRIM.

### 2.2.2 TRIM

TRIM solves the hydrodynamic equations on a Cartesian grid, allowing for coastal regions falling dry. Casulli and Stelling (1998) provide a description of the numerical implementation, extensions with regard to parallelization and nesting can be





found in Kapitza (2008). After three refinements nested oneway into a coarse grid with 12.8 km resolution covering the north-eastern Atlantic, North Sea and Baltic Sea, resolution in the German Bight is 1.6 km. The FES2004 tidal model (Lyard et al., 2006) is used to determine tidal signals at the lateral boundaries of the outer coarse grid. Wind and sea level pressure are taken from COSMO-CLM hindcasts (Geyer, 2014), which resulted from a regionalization of global NCEP/NCAR Reanalysis 1 data

(Kistler et al., 2001) using a spectral nudging technique (von Storch et al., 2000). Wind stress was parametrized according to Smith and Banke (1975), a parametrization validated from gentle breeze to gale force winds.

### 2.2.3 Effects of winds and waves

Eulerian marine currents can usually not fully account for observed drifter movements. Additional wind effects may manifest themselves in different ways. This study explores the strengths of windage effects and Stokes drift as alternative tuning

parameters for optimizing simulated drift trajectories.

Hourly fields of surface Stokes drift were simulated with the third generation spectral wave model WAM (WAMDI-Group, 1988; Komen et al., 1996), extending an existing wind-wave hindcast for the years 1949-2014 (Groll and Weisse, 2016) and including surface Stokes drift as a new element of archived model output. Wave simulations were driven with the same COSMO-CLM hindcast also used for simulations with TRIM. The wave model was used in a nested mode, with the finer

spatial resolution of about $3 \times 3$ nautical miles over the entire North Sea. Wave breaking and depth refraction were enabled. For the present study, an explicit assumption about the vertical profile of Stokes drift (Breivik et al., 2016, for instance) was replaced by a simple weighting factor $\alpha$ in Eq. (1). Choosing $\alpha = 0.5$ resulted in a reasonable fit with observations.

Windage (or leeway) effects occur when a drag resulting from parts of a drifter being exposed to the wind is not fully compensated by a drogue attached to the drifter. Generally, the direct influence of winds on the drifter type used in this

experiment is supposed to be small as long as the drogues attached are in a proper condition. However, specification of windage effects may also be needed when marine currents used do not adequately represent the surface layer drifters are immersed in. An extra wind drift parametrised as 0.6 % of 10 m wind velocity was used in combination with archived BSHcmod currents averaged over a 5 m depth surface layer. By contrast, drift simulations based on TRIM output (1 m depth surface layer) were performed without taking into account additional wind effects.

### 2.2.4 Analysis of 25 h drifter displacements

Comparing simulated trajectories with concurrent observations enables a qualitative assessment of a model's ability to reproduce overall drift patterns. However, accumulation of possibly intermittent simulation errors makes it difficult to localize the origin of major deviations in either space or time. Therefore, a series of short term (25 h) simulations was started once per day (13:00 UTC) from each drifter's observed location at that time. The short term simulation errors were analysed against the

backdrop of prevailing winds and residual currents (cf. Sect. 2.3).



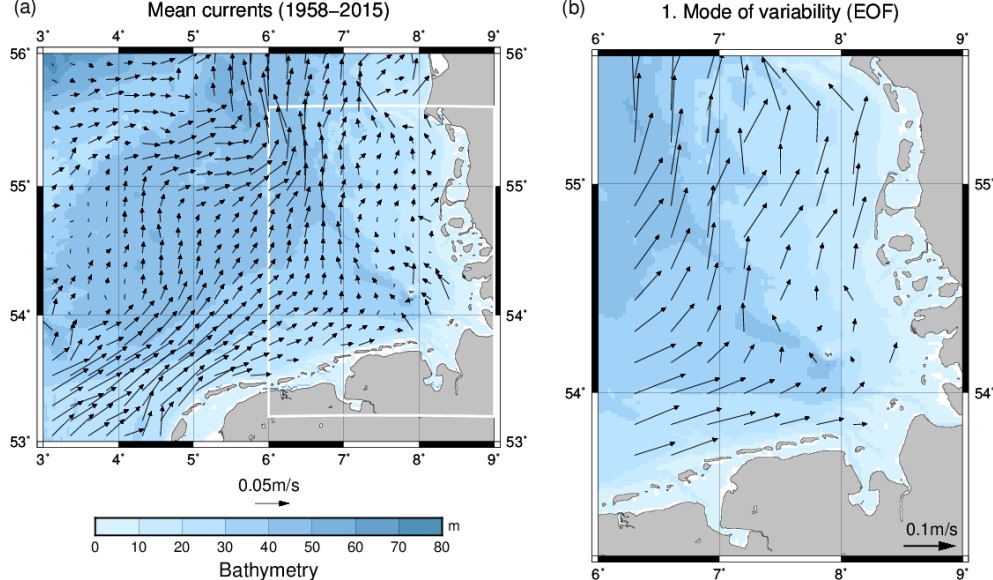

**Figure 2.** (a) Mean currents in the inner German Bight, calculated running a 2D-version of model TRIM for the period Jan 2014 - Aug 2015. (b) Leading mode of variability (1. EOF, cf. von Storch and Zwiers (1999)) of daily 25 h mean currents obtained from a PCA restricted to data from the white box region in panel (a) (Callies et al., 2017). Vector densities in the plots do not represent spatial resolution of the underlying model (1.6 km). Vectors in the right panel are scaled in such a way that the EOF represents an anomaly that would arise from the first principal component ($PC_1$) assuming the (positive) value of one standard deviation.

## 2.3 Characterization of residual currents on a daily basis

BSH classifies the residual circulation in the German Bight (between $53.25°$ N and $55.5°$ N and between $6.5°$ E and $9.0°$ E) on a daily basis, referring to surface currents from the operational model BSHcmod.[1] The classification is performed manually based on subjective assessments of 24 h averages. The small deviation of the averaging interval from two tidal periods does not affect the analysed frequency distribution of circulation patterns. Most frequent are a cyclonic circulation with a pronounced inflow at the south-western border and outflow at the northern border, a reverse anticyclonic circulation, and a category with variable current patterns. Cyclonic circulations correspond with what is observed in the long-term mean (cf. Fig. 2(a)). Six specific directional types with currents towards the E, W, N, S, NW, and SE play only minor roles. They are related to strong local winds and for statistical purposes combined into just one class. Due to topographical constraints, SW and NE patterns do not occur. Fig. 4 includes results of the BSH classifications for the period relevant in this study.

An alternative analysis is based on the 2D-version of TRIM. Slightly different from the above approach, Callies et al. (2017) defined residual currents as 25 h means (close to one lunar day = 24.8 h). These fields were than subjected to principal component analysis (PCA), focusing on the inner German Bight (east of $6.0°$ E and south of $55.6°$ N, see Fig. 2) and excluding

---

[1]http://www.bsh.de/de/Meeresdaten/Beobachtungen/Zirkulationskalender_Deutsche_Bucht/index.js



inshore areas with a bathymetric depth of below 10 m. Corresponding data are freely accessible at Callies (2016). Figure 2(b) displays the leading mode of variability (first Empirical Orthogonal Function (EOF), cf. von Storch and Zwiers (1999)). The time series of corresponding principal component $PC_1$ is shown in Fig. 4. The interpretation of the dominant anomaly pattern $EOF_1$ (it explains more than 70% of residual current variability in the area) is a strengthening or weakening (even reversal) of

mean residual currents (white box in Fig. 2(a)), depending on whether values of $PC_1$ are positive or negative.

## 3  Results

### 3.1  Observations

Figure 3 shows six observed drifter trajectories, disregarding the tracks of drifters #2, #3 and #4 that were recorded for just a view days. To facilitate a synopsis of synchronous drifter movements, the time coordinate was subjectively segmented,

assigning different colours to periods with different drift behaviour. In this context, a continuous daily index was introduced, counting days since when the first 25 h simulation for drifter #5 was started on May 27 at 13:00 UTC (cf. Table S1).

Figure 4 places drifter schedules into the context of variable atmospheric winds and marine residual currents. For each drifter, a segmented time bar covers all 25 h simulations started every day at 13:00 UTC (cf. Sect. 2.2.4). Time bars slightly truncate the times drifters were tracked, due to a disregard of hours before 13:00 on the day a drifter was deployed and of

those 25 h simulations that would extend beyond existing observations. To represent atmospheric forcing used in BSHcmod and TRIM, respectively, simulated 10 m winds at location 55° N and 7° E near the centre of the study area are shown together with observations on the island of Helgoland (54.10° N / 7.53° E). All wind vectors represent 25 h means and are plotted at the center of the respective 25 h interval starting at 13:00 UTC. The different data sources are in reasonable agreement with each other.

Figure 4 also includes the representation of a) the subjective classification of daily mean BSHcmod surface currents and b) the first principal component ($PC_1$) of 25 h mean currents simulated with a 2D version of TRIM (cf. Sect. 2.3). Positive values of $PC_1$ (or amplitudes of the anomaly pattern shown in Fig. 2(b)) indicate a strengthening of the mean cyclonic circulation, negative values refer to its weakening or even reversal. Although the two representations of residual current variability have different roots (different models, surface layer vs. vertical means, subjective vs. objective, different atmospheric forcing),

a clear correspondence between the two representations is discernible. While cyclonic hydrodynamic regimes and positive values of $PC_1$ tend to coincide with winds from the southwest, anticyclonic circulations and negative $PC_1$-values are mainly driven by winds from the northwest (Callies et al., 2017).

Concerning drifters #6 and #8, an interesting special situation occurs during June 7-16 (or days 11-20). When drifter #8 was deployed (May 30, day 3), drifter #6 already travelled for nearly 3 days and was located at a distance of about 20 km.

Within the next 4 days the two drifters further separated. On June 4 (day 8), however, they suddenly started converging quickly. From June 8 (day 12) onwards, drifters #6 and #8 stayed at a distance of less than 2 km for nearly 10 days. Distance between the two drifters as a function of time is shown in Fig. 5. Just after the distance had reached its minimum (about 800 m), the drifters started to separate again. Other short periods of fast convergence occurred later but never again the two drifters came







**Figure 3.** Observed trajectories of six drifters deployed at the locations indicated by black crosses. Drift paths were segmented using the colour code introduced in Table S1. The numerical data underlying this plot can be found in the supplementary material.



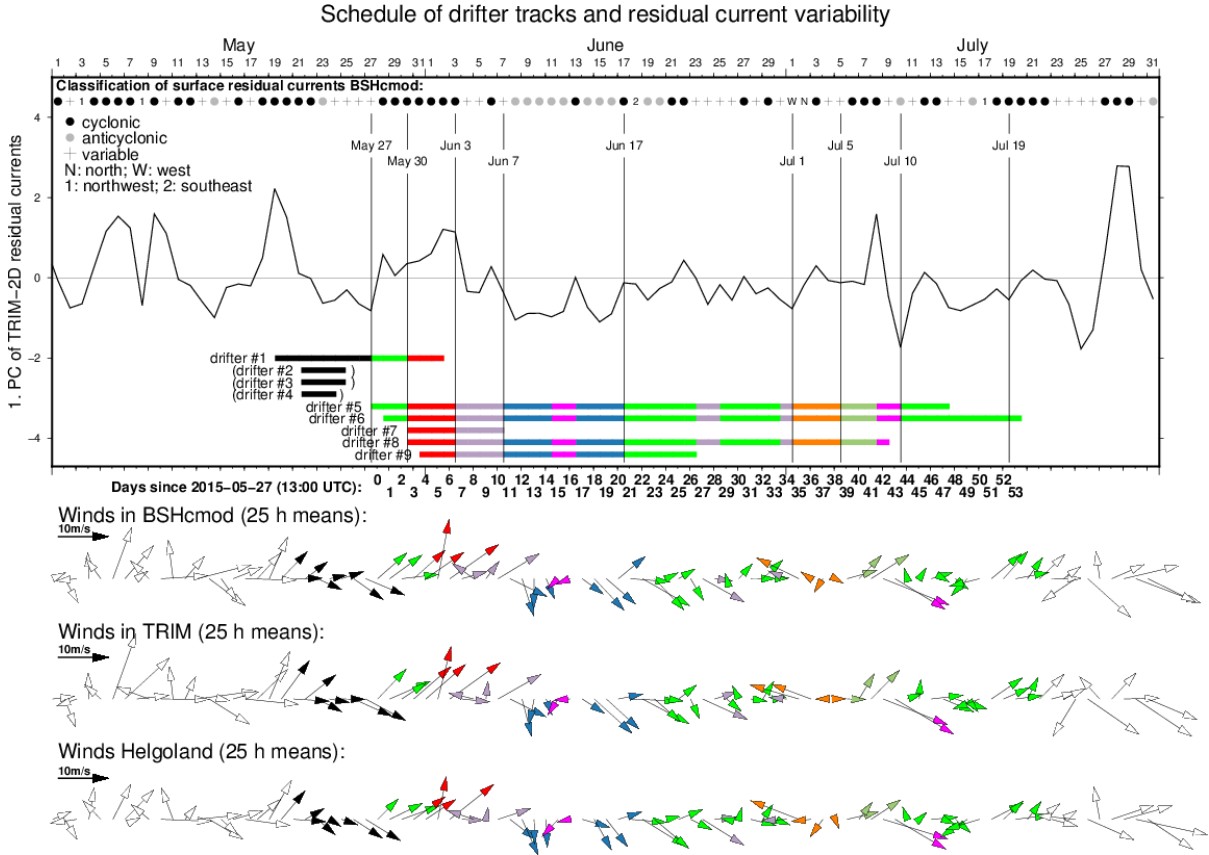

**Figure 4.** Time bars indicate for each drifter the period for which corresponding simulations were performed (drifters #2, #3 and #4 were disregarded in this study). The colour code defined in Table S1 was used for time segmentation. Symbols at the top represent the classification of daily surface residual currents based on BSHcmod. In addition the time series of the leading principal component ($PC_1$) of 25 h mean currents simulated with TRIM-2D is shown (see Sect. 2.3). $PC_1$ values were normalized with their standard deviation during the years 1958-2015. Positive $PC_1$ values represent a strengthening of the cyclonic regime, negative values its weakening or even reversal (cf. Fig. 2(b)). 25 h mean wind vectors (10 m height) used in the two model systems (extracted for location $55°$ N and $7°$ E) are contrasted with observations on the island of Helgoland ($54.10°$ N / $7.53°$ E).





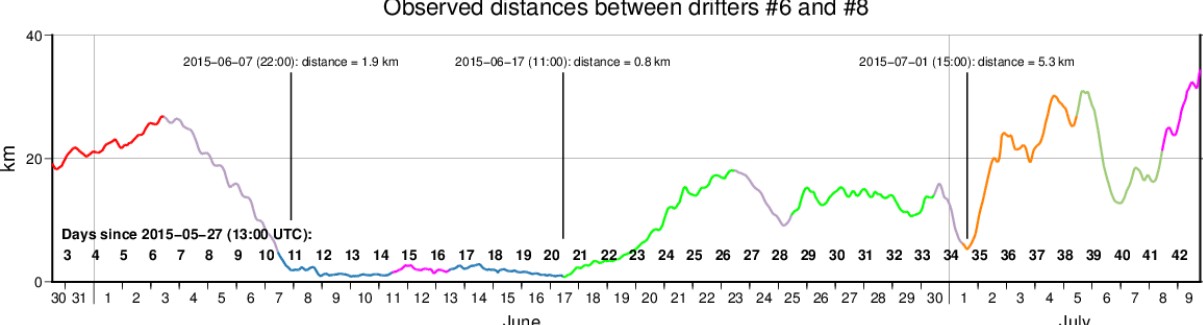

**Figure 5.** Observed distances between drifters #6 and #8. Colours refer to those specified in Table S1.

that close. During the last 8 days of their joint journey (starting at around day 35) the distance between the two drifters showed particularly large oscillations.

According to Fig. 3, a feature shared by at least four drifters (#5, #6, #7 and #8) is a general displacement towards the northeast. This movement is particularly fast in the beginning (days 3-6), brought about by persistent south-westerly winds and

a corresponding cyclonic circulation at this time (Fig. 4). Other periods with particularly fast movements occur around day 35 and days 42-43. In the former case, strong winds from the southeast trigger a very fast separation of drifters #6 and #8 (cf. Fig. 5). In the latter case, north-westerly winds give rise to extreme drift speeds in south-east direction. Drifters #5 and #8 are already in nearshore areas at that time (Fig. 3).

In their central parts, drifter trajectories #5, #6 and #8 exhibit variable drift directions but mostly moderate drift velocities.

Although structures are complex, they show resemblance, explicable by moderate distances between the three drifters. Drifter #9, started further away in the south-east of the domain, behaves more differently, although some structural coherence with other drift paths can still be distinguished. In particular this applies for days 11-20 (June 7-16) characterized by the close proximity of drifters #6 and #8. The journey of drifter #1 has just a small overlap with those of other drifters, the drifter is soon trapped within the entrance to tidal basins.

Further details of observed trajectories will be addressed in Sect. 3.2 together with a presentation of corresponding simulations.

## 3.2   Simulations

Taking drifter #5 as an example, Fig. 6 compares the simulation based on TRIM with three different simulations based on BSHcmod. The three different setups are a) just Eulerian currents (BSHcmod), b) Eulerian currents plus windage (BSHc-

mod+W, windage parametrized as 0.6 % of 10 m winds) or c) Eulerian currents plus 50 % of surface Stokes drift simulated independently with wave model WAM (BSHcmod+S). Corresponding simulations for all six drifters can be found in the appendix (Figs. A1-A4). Numerical data displayed in the graphs are provided as supplementary material. It appears from Fig. 6 that combining BSHcmod currents with either windage or Stokes drift brings corresponding simulations closer to those based





on Eulerian surface currents from TRIM. A key effect of the inclusion of extra wind or wave effects is the intensification of westward transports in agreement with wind directions that occur most frequently.

A more detailed assessment of model performance is enabled by analysing drifter displacements on a daily basis. Simulations of 25 h drift paths (cf. Sect. 2.2.4) were initialized on every day 0-53 (cf. Table S1) at 13:00 UTC. Full sets of corresponding

plots are provided as supplementary material, referring to the different model setups also shown in Fig. 6. Each of four sets of plots (SM1-SM4) contains information for all days 0-53. The first collection (SM1) shows 25 h drifter displacements that were observed. The second (SM2) compares corresponding simulations based on either BSHcmod or TRIM Eulerian currents with these observations. The third (SM3) is similar except that BSHcmod currents are complemented by parametrized windage (BSHcmod+W). Finally, the fourth collection (SM4) compares two BSHcmod simulations including either windage

(BSHcmod+W) or Stokes drift (BSHcmod+S).

Drawing on the material from SM3, Figs. 7 and 8 present results for eight selected days, comparing simulations based on TRIM surface currents with those based on BSHcmod+W. Each panel combines all drifters that are available at the respective time. Observed drifter displacements are coloured in agreement with Table S1.

Concentrating on the four drifters that travelled longest, histograms in Fig. 9(a) show daily values of separations between

observed and simulated end points of 25 h drift paths, referring to simulations with either TRIM or BSHcmod+W. To show the relative importance of drift errors, total distances covered according to observations or simulations are also included. Figure 9(d) shows the angles between observed and simulated drifter displacements. Time series (25 h means) of wind speeds used in TRIM and BSHcmod (and also BSHcmod+W) are shown in Fig. 9(b) together with surface Stokes drifts from wave model WAM. Figure 9(c) copies observed Helgoland wind vectors from Fig. 4.

The following description highlights some key aspects of drifter observations and concurrent simulations during different sub-periods of the experiment. The description focusses on simulations based on either BSHcmod+W or TRIM (cf. SM3 for the full set of results).

**Days 0-6 (27 May - 2 June):** The period is characterized by cyclonic residual currents increasing in strength (Fig. 4). Driven by winds mainly from southwest, drifters move fast towards a north-eastern sector. After about one week, simulated

locations of drifter #5 (Fig. 6) and drifters #6, #7 and #8 (Figs. A2 and A4) are in reasonable agreement with observations (Fig. 3). The 25 h drifter displacements reveal that by and large both models capture observed drift speeds reasonably well (Fig. 9(a)). Although for both models simulated drift distances agree well with observations, particularly for TRIM the errors of simulated trajectory end points can be appreciable in terms of absolute values (Fig. 9(a)). However, these errors must be assessed in the light of total distances covered. According to Fig. 7(f), errors arise mainly from moderate

directional deviations in combination with large displacements. In most cases simulations are rotated to the right relative to observations. Directional mismatches tend to be larger in TRIM than in BSHcmod+W. In both models, they are largest for drifter #9 on day 4 (Fig. 9(d) and SM3).

While on day 2 the neighbouring drifters #5 and #6 move into substantially different directions, both models predict parallel movements in a direction close to an average of the two observations (SM3). In Fig. 9(d) this leads to directional



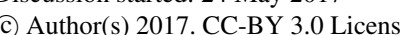

**Figure 6.** Top: trajectories of drifter #5, simulated based on currents from (a) BSHcmod and (b) TRIM, disregarding extra effects of winds or waves. Bottom: Simulations based on BSHcmod including either (c) windage (BSHcmod+W) or (d) Stokes drift (BSHcmod+S). Corresponding observations are shown in Fig. 3(b). Figs. A1-A4 in the appendix show corresponding simulations also for all other drifters. Underlying numerical data are also provided as supplementary material.

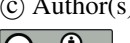





**Figure 7.** Observed 25 h drift paths (coloured in agreement with Table S1) are contrasted with concurrent simulations (black) based on BSHcmod+W (a-d) or TRIM (e-h). For four selected days, panels combine all drifters observed at the time of the plot. All drift distances were converted into 25 h mean drift velocities. Note that the length scales shown do not correspond with the spatial scale of the geographic map. Vectors in each panel's top right corner indicate the mean wind velocity vector at 55° N and 7° E derived from the respective atmospheric model used.

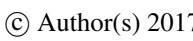



**Figure 8.** Continued from Fig. 7.



errors of opposite sign. On day 3 (Figs. 7(a) and 7(e)), both models simulate parallel displacements of drifters #5, #6 and #8 quite well, but only BSHcmod+W captures the deviant direction of drifter #7. Although the agreement with observations depends on the inclusion of windage, simulated differences in directions clearly arise from information in BSHcmod Eulerian currents. By contrast, on day 6 (Figs. 7(b) and 7(f)) the again deviant direction of drifter #7 (now rotated to the other side) is no longer reproduced by BSHcmod+W.

For drifter #1 simulations are generally poor. On day 0 both models well reproduce its drift direction but much underestimate its drift speed (SM3). On days 3-5, drifter #1 already enters the complex coastal bathymetry which is insufficiently resolved in both models (e.g. Figs. 7(a) or 7(e)).

**Days 7-10 (3-6 June):** The strong cyclonic regime declines. While the residual current classification based on BSHcmod is inconclusive, values of $PC_1$ tend to change from positive to negative values (Fig. 4). Strong south-west winds first cease and then blow from different directions with different strengths. After a steep decline, observed displacements of drifters #5, #6 and #8 take a minimum on days 7 or 8 (Fig. 9(a)). On day 7 directional errors of TRIM simulations are particularly large and mostly opposite to those based on BSHcmod+W (Fig. 9(d)). On the following days both observed and simulated drift lengths start increasing again.

According to observations, drifter #9 is the only one that rotates its movement from north-east to north-west already on day 7, all other drifters follow on day 8 (SM3). According to Fig. 9(a), observed speeds of drifter #9 peak on day 8, when drift speeds of drifters #5 and #6 take a minimum (see also Figs. 7(c) or 7(g)). Observed drifter displacements seem to decrease with distance from the coast, a variation not resolved in any of the two simulations. Simulations based on TRIM currents show for all drifters #5, #6, #8 and #9 larger displacements than BSHcmod+W simulations. But while displacements according to TRIM are still too small for the two nearshore drifters #7 and #9, displacements of more offshore drifters #5 and #6 are largely overestimated (Fig. 7(g)). Neither simulated currents nor fields of windage or Stokes drift are able to reproduce the spatial gradients.

Observed sub-mesoscale differences in drift speed (e.g. day 8, Fig. 7(c)) and direction (e.g. day 10, Fig. 7(d)) are also the reason for the fast convergence of drifters #6 and #8 (Fig. 5) that both models are unable to resolve. However, it must also be noted that the more eastern direction of the movement observed for drifter #6 on day 10 (Fig. 7(d)) is not shared by any of the neighbouring drifters moving more to the north-east.

Neither model captures the different behaviour of drifter #7 (compare Figs. A2(d) and A4(d) with Fig. 3(d)). Beginning at about day 8 (Fig. 7(c)), drifter #7, deployed at a more northern location, continues its fast movement towards northern directions, while all more southern drifters considerably slow down.

**Days 11-14 (7-10 June):** Winds from the north-west or north trigger an anticyclonic circulation (Fig. 4). On day 11, the inclusion of windage much reduces errors of BSHcmod simulations for drifters #6, #8 and #9 (compare panels in SM2 and SM3), mainly due to improved drift directions. Only for drifter #5, moving much slower despite its vicinity to



other drifters, adding windage leads to a drift velocity that is much overestimated. This overestimation also holds for simulations based on TRIM Eulerian currents (cf. Fig. 9(a)).

Also during days 12-14, adding windage impairs the overall quality of BSHcmod drift simulations due to modified drift directions, despite the fact that drift speeds are improved. This seems to be the main reason why TRIM simulations tend to perform better on these three days (Fig. 9(a)).

Note, that after about day 12 both observations and simulations for the two drifters #6 and #8 are more or less plotted on top of each other and can therefore hardly be distinguished (applies to Figs. 8(a,e), for instance, and various plots in the supplementary material).

**Days 15-16 (11-12 June):** A short period with low winds from the north-east or east during which the circulation returns to a cyclonic orientation (Fig. 4). In particular on day 16, BSHcmod+W simulations much underestimate observed drift speeds of all drifters (Fig. 9(a) and SM3). Due to the low wind conditions, adding windage or Stokes drift (Fig. 9(b)) cannot eradicate this deficiency which is much less pronounced in TRIM simulations (see Fig. 9(a)).

In Fig. 3, excursions towards the north-west can be discerned for drifters #5, #6, #8 and #9 during days 15-16. For drifter #5, Fig. 6(b) shows how TRIM just adumbrates such movement while all transports simulated based on BSHcmod more or less cease for the two day period.

**Days 17-20 (13-16 June):** During days 17-19 the circulation returns to a anticyclonic orientation, driven by north-westerly winds (Fig. 4). On those days, simulations based on either model consistently produce drift velocities that are markedly rotated to the left of corresponding observations (cf. Fig. 9(d) and Figs. 8(a) and 8(e)).

On day 20 the wind direction returns to the south-west in BSHcmod or west in TRIM (cf. Fig. 4), the residual circulation becomes cyclonic for one day. Under these transitional conditions, signs of directional errors substantially differ between the two models (Fig. 9(d) and SM3).

Comparing TRIM simulations for drifter #5 (Fig. 6(b)) with corresponding observations (Fig. 3(b)) reveals that TRIM currents imply too strong southward transports. This finding holds also for TRIM simulations of drifters #6 and #8 (Fig. A4). TRIM drift speeds being higher than their observed counterparts can also be discerned from Fig. 9(a).

**Days 21-26 (17-22 June):** A mainly anticyclonic circulation gradually changes into a cyclonic circulation towards the end of the period (Fig. 4). During the first three days (days 21-23) considerable errors in both simulations resemble each other to a surprising degree (SM3). In particular this applies to drift directions that are typically rotated to the left of observed counterparts (Fig. 9(d)). An exception forms drifter #9, for which tracks simulated with either model change from eastward (day 21) to southward (day 23) while in reality drifter #9 constantly moves towards the southeast (see also SM3).

On day 24 both model simulations are particularly poor in that they much underestimate observed drift velocities (Fig. 9(a)) during a transition from anticyclonic to cyclonic residual currents (Fig. 4). Simulated drift angles now differ considerably between the two models (Fig. 9(d)), although during low drift speeds already mentioned. An extreme





misfit of simulations occurs for drifter #9. However, already from day 22 onward observed displacements of drifter #9 are much larger than their simulated counterparts (Fig. 9(a)). The reason for this remains unclear. Fig. 3(f) illustrates the fast movements of drifter #9, mostly in line with prevailing wind directions. It cannot be precluded that drifter #9 experienced some problem with its drogue. The special role of drifter #9 is most pronounced on day 26. While other

drifters stop moving according to both observations and simulations, drifter #9 does hardly slow down.

From about day 22 onward drifters #6 and #8 start separating again (Fig. 5 and SM3). On day 22 drift speeds differ, on day 23 also drift directions. As to be expected, neither model reproduces such sub-grid scale differences observed.

**Days 27-28 (23-24 June):** High wind speeds from the north-west on day 27 give rise to southern transports. Success of corresponding BSHcmod+W simulations (Fig. 8(b)) much depends on the inclusion of windage, which produces a more

realistic drift speed. However, at the same time pushing drifter displacements more towards wind direction increases errors in drift direction (compare SM2 and SM3). Simulations based on TRIM are more consistent even without windage (Fig. 8(f)).

Next day (day 28) winds abate and marine surface transports veer to a northeast direction. Model simulations seem to lag behind this sudden change. But there are also substantial differences between observed speeds of the neighbouring

drifters #6 and #8, which implies a short period of their fast convergence (Fig. 5).

**Days 29-33 (25-29 June):** A period with variable wind directions. Although wind speeds are not always small (not shown), strengths of 25 h averages are generally low (Fig. 9(b)) due to compensating effects on the short time scale. Model performances are poor with major errors in both directions and drift speeds. Drifter displacements are generally underestimated (Fig. 9(a)). Observed northward transports (e.g. for drifter #8, see Fig. 3(e)) are not reasonably reproduced by

simulations based on BSHcmod+W (Fig. A2(e)) and even less in simulations based on TRIM (Fig. A4(e)). Note, however, that simulations shown in the latter figures are not exactly consistent with 25 h simulations started from observed drifter locations.

**Day 34 (30 June):** A day with a fast convergence of drifters #6 and #8 (Fig. 5). This convergence is caused by the fact that drifter #8 shares with drifters #5 and #6 neither direction nor speed of its very fast west-northwest movement (SM3). No

model resolves the substantial differences observed.

**Days 35-38 (1-4 July):** A four day period in which drifters #5, #6 and #8 all move quickly into northern or northwestern directions (Fig. 3). The by far largest drifter displacements occur on the first day (day 35, cf. Figs. 8(c) and 8(g)) with strong winds from the southeast. Drift directions are consistent with the BSHcmod classification of residual currents on 1-2 July as being directed towards the north and west, respectively (Fig. 4). Note that the counts of days used in this

study are based on 24 h intervals starting at 13:00 UTC, i.e. they are shifted by about 12 h relative to the calendar days underlying the BSHcmod classification.

As already on day 34, a particularly fast movement of drifter #8 is also observed on day 35. Drifter #8 also drifts more westwards than drifters #5 and #6. This special behaviour underlies the very quick separation of drifters #6 and #8 at that





time (Fig. 5). The drift direction of drifter #8 is more aligned with wind direction than those of its companion drifters, possibly indicating problems with the drogue.

On day 36, TRIM (but not BSHcmod) assumes the wind to persist (Fig. 4 or SM3), which results in a substantial overestimation of drifter displacements (Fig. 9(a)). A comparison with observations at Helgoland (Fig. 4) suggests that winds used by BSHcmod are more realistic. Regarding drift directions, simulations based on either model are rotated to the left relative to observations.

On days 37-38 both models assume low 25 h mean winds (Fig. 9(b)) although on an hourly basis some short term peaks exist (not shown). For all drifters, simulations based on either model are very poor. BSHcmod+W simulations predict drifter movements to cease, which contradicts observations on day 37 (SM3). According to TRIM, drifters keep moving (possibly brought about by the assumed stronger winds the day before), but predicted drift directions much deviate from observations. On day 38, drifters #6 and #8 coming to rest is well reproduced in both models. Drifter #5, however, continues to move, a special behaviour that remains unresolved in both models.

**Days 39-41 (5-7 July):** Freshening southwesterly winds bring about a transition towards a strengthened cyclonic circulation (Fig. 4). A remarkable observation is the extremely fast movement of drifter #8 in reaction to this forcing (Fig. 3(e)). One side effect of this special behaviour is the rapid reduction of the distance to drifter #6 during day 39 (Fig. 5). While simulations for drifters #5 and #6 perform well, the behaviour of drifter #8 cannot be reproduced.

**Days 42-43 (8-9 July):** The wind veering from southwest to northwest implies a fast transition from a cyclonic to an anticyclonic residual current regime (Fig. 4). Models perform well for drifters #6 and also #5, but again simulations for drifter #8 are very poor on day 42 (Figs. 8(d) and 8(h)), the last day on which drifter #8 data are available. On day 43 the fast southward movements of drifters #5 and #6 are reasonably well represented in both models (SM3).

**Days 44-53 (10-19 July):** In this final period only drifters #5 and #6 are left, both of them already located in coastal waters. Wind speeds are mostly low. Directional drift errors tend to be large and in particular at the end of the period speeds of drifter #6 are in most cases underestimated by model simulations (Fig. 9(a)). Between about day 45 and day 48 extra large differences between averages of wind velocities used in BSHcmod and TRIM can be discerned in Fig. 9(b). Consequences of a sudden reversal of the mean wind direction between day 50 and day 51 can reasonably be represented in both models.

## 4 Discussion

Incorporating either direct wind drag or Stokes drift proved indispensable for successful surface drifter simulations based on BSHcmod currents. The currents representative for a 5 m depth surface layer, integrated and archived from original BSHcmod output, could not properly represent a direct influence of winds, possibly mediated via waves. Based on TRIM currents representative for a surface layer of only 1 m depth, drift velocities were generally larger and similar in size to those observed





**Figure 9.** (a) Distances between observed end points of 25 h drift paths and corresponding simulations based on either BSHcmod+W or TRIM. All drift errors, coloured and labelled in terms of days since May 27 (13:00 UTC), are assigned to the center of the respective 25 h period. In addition, the panel shows total distances travelled. (b) Wind speeds used in the two models and surface Stokes drifts obtained from wave model WAM. All these data were extracted for the central example location 55° N and 7° E. (c) Helgoland winds, copied form Fig. 4. (d) Angles between observed and simulated tracer displacements. Throughout the figure, all values represent 25 h averages.



(Fig. 9(a)). In many cases, however, 25 h simulations based on BSHcmod+W outperformed those based on TRIM, while in other cases (e.g. days 13-16) TRIM simulations were still in better agreement with observations (Fig. 9).

In several other studies (e.g. Gästgifvars et al., 2006; Kjellsson and Döös, 2012; De Dominicis et al., 2012) simulated marine surface currents were found being too small, possibly also due to insufficient resolution of the marine surface layer. As a side
effect, predictions may be particularly good when marine currents and winds are nearly parallel (Gästgifvars et al., 2006). The drift component most underestimated based on just BSHcmod Eulerian currents was a displacement towards the northeast, along the most frequent wind directions (compare Figs. 3(b) and 6(a)). This deficiency could very effectively be remedied by adding direct effects of winds or waves. However, during periods when anticyclonic residual currents prevail (along with winds from the north-west, for instance), currents will generally not parallel winds (e.g. day 18, Fig. 8(a)), unlike the situation
with south-westerly winds driving a cyclonic circulation (e.g. day 3, Fig. 7(a)). Erroneous residual surface currents in the inner German Bight can therefore not always be fixed by simply adding windage or Stokes drift.

Drifters #5, #6 and #8 played a central role in this study because their trajectories overlapped for 40 days, enabling tentative conclusions regarding spatial scales that affected long- and short-term drifter displacements. Wind fields resolved in numerical models (and also corresponding fields of Stokes drift) tend to vary smoothly on a regional scale. A substantial impact of
winds on surface currents may be one aspect contributing to the fact that simulated trajectories resemble each other more than corresponding observations. But also the observed drifter paths show similarities that point to the impact of large scale forcing.

Due to bathymetric constraints and different scales of relevant processes, spatial variability of marine currents tends to be higher than that of wind fields (Röhrs et al., 2012). However, our study did not show clear effects of the higher resolution in BSHcmod regarding either space (900 m compared to 1.6 km in TRIM) or time (15 min compared to 1 h in TRIM). Like
TRIM, also BSHcmod proves itself unable to reproduce the specific behaviour of drifter #7 during days 7-11, for instance (Fig. 3(d)). This could suggest that some relevant aspects of nearshore transports are not properly represented in both models. Surprisingly small effects of resolutions in both space and time on the metrics for Lagrangian predictability were also reported by Huntley et al. (2011).

Drifters will separate even if they start together from about the same location. Ohlmann et al. (2012) performed an experiment
with initial distances of 5-10 m between drifters so as to possibly better resolve an initial phase of so-called *non-local* dispersion characterized by exponential growth of the mean square pair separation, driven by eddies larger than the distance between the two drifters. In the present field experiment, simultaneous deployments of drifters #2 and #3 were originally intended to study an example of drifter dispersion. The two drifters, both tracked over 3.7 days, stayed very close together for some time until they abruptly started to separate. There were doubts, however, that this separation might have been triggered by an unobserved
interaction with the research vessel. As the drifters crossed wind parks, it could also be that they had interfered with a turbulent wake related to the pile of an engine. Due to such concerns, drifters #2 and #3 were excluded from the present analysis.

Fortunately, drifters #6 and #8 offered another opportunity to estimate predictability of drift trajectories. The minimum distance of only 800 m qualified the two drifters as a 'chance pair' (e.g. (Döös et al., 2011)). Note that drifters #6 and #8 were of different types (cf. Tab. 1), which could imply accelerated spatial separation. On the other hand, the two drifters
travelling jointly for about 10 days in a sense justifies the assumption that consequences of different designs were not essential.





Also Fig. 9(a) provides no evidence for major differences in the overall behaviours of drifters #6 and #8 during the period of interest.

From the perspective of a model with either 900 m (BSHcmod) or 1.6 km (TRIM) grid resolution, the locations of drifters #6 and #8 almost coincided for about 10 days (cf. Fig. 5). Thereafter the two drifters' separation increasing by a rate of about

3 km per day (according to visual inspection of Fig. 5) indicates a lower bound of prediction uncertainty under these specific conditions. An independent second estimate can be obtained considering the period when the two drifters converged (days 8-11). Assume that modelling were undertaken to determine where an item collected on day 11 came from. Looking 4 days back in time, the two drifters #6 and #8 have separated by about 20 km, so that the uncertainty estimate (about 5 km per day) even exceeds the above value. However, the separation rate is still much lower than that reported by Huntley et al. (2011, their Fig. 3)

under open ocean conditions near the Kuroshio current, considering a similar constellation with two drifters that separate after staying close for a couple of days. A wide spectrum of typical separation rates in different regions worldwide provided by Barron et al. (2007) also shows systematically larger values.

Error bounds estimated from drifter convergence/divergence will combine with model deficiencies that at least theoretically could be eliminated by model improvement or calibration. However, the above error estimates roughly fit into the general range

of simulation errors found in this study (Fig. 9(a)). Ohlmann et al. (2012) tried to reproduce observed drifter trajectories with a Lagrangian stochastic model based on Eulerian background velocities derived from HF-radar observations interpolated to a regular $2 \times 2 \, \text{km}^2$ grid. Substantial discrepancies exceeding the expected level of HF radar measurement errors were found in occasional periods. On average, the separation between corresponding centres of gravity was found to be about 5 km after 24 h, a value that compares well with uncertainties estimated from the present experiment. It remains as an open question whether

the quality of predictions would be better with HF-radar observations replacing output from numerical models.

According to Koszalka et al. (2009) and Döös et al. (2011), 'chance pairs' should possibly be distinguished from pairs of drifters intentionally launched together, because their behaviour may depend on specific hydrodynamic conditions. An interesting question is what characterizes the 10 day period when drifters #6 and #8 stayed close together. The drifter convergence (days 7-10) coincided with the transition from a cyclonic to an anticyclonic residual current circulation (Fig. 4). The anticy-

clonic regime forced by winds from mainly the northwest dominated days 11-20, except for a short episode (days 14-16) with very low winds and a circulation returning to the cyclonic orientation for about one day. Drifters #6 and #8 started separating again when residual currents gradually returned to an either indifferent or cyclonic circulation, a process probably best represented in the time series of $PC_1$ in Fig. 4. So it seems that both convergence and divergence of the two drifters coincide with reorientations of the hydrodynamic regime.

The present data are insufficient for a discussion of to which extent the drifters' observed responses to changing winds and residual currents depend on drifter location. The small number of drifters can obviously not represent the spatial structure of transports. Based on model simulations, however, there are promising techniques to better describe regions within which separation for drifters can be expected. Identification of Lagrangian coherent structures (LCS) is a field that developed recently (e.g. Shadden et al., 2009). Huhn et al. (2012) applied LCS to identify transport barriers for drifters in an estuary, Peacock and

Haller (2013) discuss how such techniques could be used for optimizing drifter deployment in the sense of maximizing their





dispersion. Olascoaga et al. (2013) used LCS to illustrate how mesoscale circulation shapes near surface transports in the Gulf of Mexico.

In both BSHcmod+W and TRIM simulations, drifter displacements were often rotated to the left of their observed counterparts, e.g. during days 13-23 (cf. Fig. 9(d)). A parametrization of wind induced Ekman drift (Röhrs and Christensen, 2015)
might be explored as a means to remedy such model deficiencies including lacking representation of the Coriolis-Stokes drift (Hasselmann, 1970; Polton et al., 2005) driven by ocean surface waves. However, the directional bias is not permanent and it seems difficult to justify and calibrate a corresponding parametrization based on the limited amount of data from the present field study.

A couple of different processes can be relevant for an exchange of energy and momentum between surface waves and
underlying mean currents (cf. Smith, 2006). Under open sea conditions, the probably most important process affecting near-surface drifters is the Stokes drift which arises when backward motions beneath the troughs of surface gravity waves do not fully compensate forward motions beneath the crests. However, a key observation from our simulation experiments is that for surface drifters the inclusion of an explicitly simulated Stokes drift did not produce an added value compared to a simple parametrization of wind drag in terms of 10 m winds. According to Fig. 9(b), wind speeds used as forcing for either TRIM or
BSHcmod are both highly correlated with Stokes drifts calculated with wave model WAM (based on exactly the same wind hindcast also used as forcing for TRIM). This similarity agrees with results reported by Drivdal et al. (2014, their Fig. 7), for instance.

From experimental data, Röhrs et al. (2012) estimated Stokes drift to be about twice as large as effects of direct wind drag. However, as the roles of direct wind drag and Stokes drift are difficult to disentangle, we did not conduct experiments with
mixtures of the two processes. Validating modelled wave effects based on four surface drifters deployed near the Grand Banks (Newfoundland), Tang et al. (2007) considered both processes in combination. They also found simulated Stokes drift to be linearly related to wind velocities, so that it seems hardly possible to decide whether the about 21 % decrease of separation between modelled and observed trajectories after one day are really attributable to Stokes drift effects. According to Breivik and Allen (2008), the difficulty to separate Stokes drift effects from an empirically parametrized direct wind drag is a major
reason why Stokes drift is neglected even in most operational search and rescue (SAR) modelling systems, where a realistic assessment of existing uncertainties and their origin is of utmost importance.

Tang et al. (2007) found Stokes drift being about 1.5% of wind speed. This value agrees with the ratio (0.3/20) of the scales annotated on the two y-coordinates in Fig. 9(b). For low wind conditions the relative importance of Stokes drift decreases (again in agreement with the results of Tang et al. (2007)), but in these cases the overall contributions from winds and waves
are small anyway.

In particular growing young wind seas forced by local winds typically produce strong surface Stokes drifts that decline fast with depth (e.g. Röhrs et al., 2012). Breivik et al. (2016) developed an approximate method to efficiently calculate this near-surface shear, underestimated by the common assumption of a monochromatic profile. Based on these formulas, Röhrs and Christensen (2015) calculated in the context of a drifter experiment in the Barents and Norwegian Sea that an average
Stokes drift of 8.9 cm/s at the surface contrasted with an average of 3.7 cm/s at 1 m depth. For the present study we did not

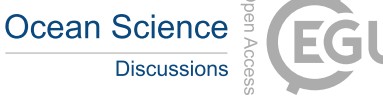



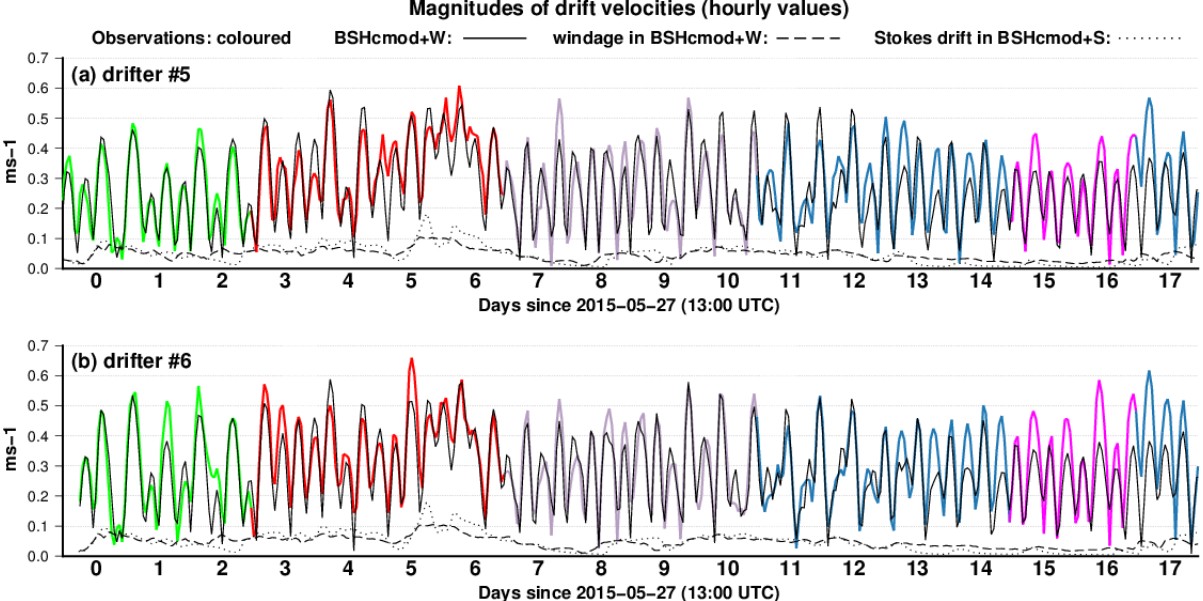

**Figure 10.** Magnitudes of drift velocities on a hourly basis, considering drifters #5 and #6. Magnitudes of observed velocity vectors (coloured) are compared with simulations based on BSHcmod+W. In addition, magnitudes of windage (in BSHcmod+W) and Stokes drift (in BSHc-mod+S) are shown. All model values are specified from either atmospheric or marine fields interpolated to observed (not simulated!) drifter locations. See supplementary material (SM5) for full times series for both drifters.

apply any theoretical profiles but just tuned a constant factor that reduces surface Stokes drift to a value suitable for the drifters deployed. The 50% factor used in BSHcmod+S simulations did not result from an in depth model calibration. Its vagueness corresponds with that of the windage factor used in BSHcmod+W. In both cases even most careful calibration would not lead to robust estimates given the limited data available.

A more general question is which relative contributions of extra wind or wave effects must be expected. Based on data from an experiment in northern Norway, Röhrs et al. (2012) found Stokes drift to be about 20 % of the mean Eulerian currents. For BSHcmod+W simulations of drifter #5 we found average magnitudes of hourly Eulerian currents to be about 0.27 m/s and corresponding values for parametrized windage (last term in Eq. (1)) about 0.043 m/s. The resulting relative magnitude of 16 % roughly agrees with the ratio which Röhrs et al. (2012) found for Stokes drift.

Figure 10 compares the magnitudes of observed and simulated drift speeds on an hourly basis, referring to trajectories of drifters #5 and #6 during days 0-17 (see supplementary material SM5 for corresponding full time series). All simulated velocity components were specified at observed rather than simulated drifter locations (i.e. no drift simulation was performed), so as to avoid the problem of spatial separation between simulations and observed counterparts. Observed and simulated drift speeds agree surprisingly well at least during about days 0-12. Nearly perfect agreement for one drifter sometimes coincides with

discrepancies for the other, a possible manifestation of sub-grid scale processes (see observations at the beginning of day 5, for





instance). Together with total drift speeds, Fig. 10 shows also magnitudes of simulated windage and Stokes drift. During most of the time, these two drift components are of similar size. More short-term pulses of Stokes drift can be discerned at days 5-6. Generally, however, contributions from both wind and waves are smooth.

In Fig. 10, both observations and simulations show regular intermittent patterns in connection with tidal cycles. Alternating heights of maximum total drift speeds indicate that movements along different branches of tidal ellipses have components that are alternately oriented in the same or opposite direction of a superimposed non-tidal drift component. This non-tidal drift is possibly but not necessarily related to wind effects. On days 13 and 14, such non-tidal drift manifests itself more in simulations than in observations, while during days 15 and 16 alternating drift speed maxima are more pronounced in observations (in particular for drifter #6). From Fig. 3(c) an observed fast drifter displacement to the northwest can be discerned during days 15 and 16. All models failed to reproduce this movement (compare Fig. 3(c) and Fig. A2(c), for instance). Considering the small values of windage and the even smaller of Stokes drift (wind directions allow for only small fetches over the open sea), tuning these effects cannot substantially improve simulations.

Figure 11 provides magnitudes of velocities of drifter #5 for the full period the drifter was tracked. However, unlike in Fig. 10(a) now magnitudes of drift velocities were calculated from moving 25 h averages of hourly vectors. Removal of compensating tidal effects enhances visibility of the contributions of wind or waves. A clear correspondence between minimum drift speeds in observations and simulations can be discerned. Note that in Fig. 11, due to vectors having different directions, separations between simulated total drift speeds and contributions of windage do not directly translate into magnitudes of mean Eulerian currents. For instance, the simulated mean speed of drifter #5 nearly vanishes on day 24 (see also SM3) as non-zero windage effects are offset by opposed Eulerian currents.

In agreement with moderating wind (Fig. 4), the overall strength of simulated windage decreases during days 11-16. The poor agreement between simulations and observations in Fig. 11 on day 11 seems surprising in the light of the good agreement in Fig. 10(a) on that day. However, the differences in Fig. 11 arise from averaging under conditions when winds veer from south-west to north-west and the hydrodynamic regime switches from a cyclonic to an anticyclonic orientation (cf. Fig. 4). Information on wind directions is not contained in Fig. 10(a). The large discrepancy between observations and simulations on days 15-16 was already addressed in the context of Fig. 10.

The period with largest differences between contributions from either windage or Stokes drift occurs during days 30-34, when indeed simulations based on BSHcmod+W and BSHcmod+S, respectively, diverge most (cf. Figs. 6(c) and 6(d)). According to Fig. 3(b) results from model version BSHcmod+W seem to be more realistic. It is interesting to see that also TRIM simulations are particularly wrong in this period, producing transports to the south-east (Fig. 6(b)) when in reality drifter #5 moved in a north-east direction (Fig. 3(b)).

Note that this study and the above discussion refer to a posterior amendment of existing simulations rather than their recalculation including all wave effects. Lacking success of this approach is not to say that deficiencies of drifter simulations are not related to wind conditions. The problem around days 15-16, for instance, occurs under non-stationary wind directions that affect also the orientation of the residual current regime (Fig. 4). Changes of wave induced forcing of the ocean, including



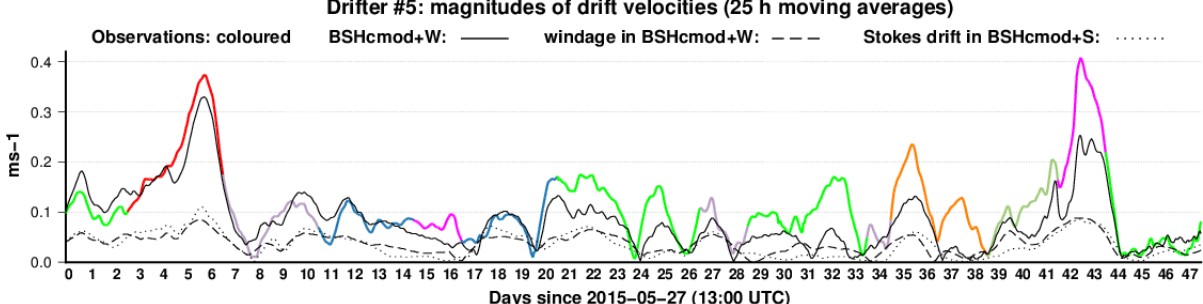

**Figure 11.** Magnitudes of 25-h moving averages of drift velocity vectors, considering drifter #5. As in Fig. 10(a), magnitudes of observed velocity vectors (coloured) are compared with simulations based on BSHcmod+W. In addition, magnitudes of windage (in BSHcmod+W) and Stokes drift (in BSHcmod+S) are shown. All model values are specified from either atmospheric or marine fields interpolated to observed (not simulated!) drifter locations.

sea-state-dependent momentum flux and Stokes drift (Staneva et al., 2016) affect water level, high and low water times and therefore also ocean currents.

Röhrs et al. (2012) warn that implementing the Stokes drift as a simple additive component of drift velocity, parameterized in terms of wind forcing, can be inconsistent (i.e. violates conservation of both momentum and energy) if Eulerian currents were
simulated without taking into account the reservoir of wave momentum and energy. Although this is the case for simulations in the present study, the exchangeability of Stokes drift and a parametrized wind drag indicates that the role of waves as a reservoir of momentum wasn't relevant at least during the period considered. One reason for this could be that due to limited fetches the North-Sea is less swell-dominated than other Nordic Seas (Semedo et al., 2015).

Two crucial questions difficult to answer are the following: a) are the drifters' behaviours really representative for surface
currents and b) can it justifiably be assumed that all drifters maintained their ideal drift properties over the whole period they were tracked. Drifter trajectories may reflect a specific exposure to winds and waves, well illustrated by the experiment reported by Röhrs et al. (2012). Edwards et al. (2006) suggested corrections to improve trajectory simulations when wind errors and characteristics of the specific drifters deployed are known. However, for the present study a tentative positive answer to the first question could be given based on the reasonable correspondence between the magnitudes of observed tracer displacements
and their counterparts simulated based on just TRIM Eulerian surface currents (cf. Fig. 9(a)). On the other hand, Poulain et al. (2009) estimated a higher downwind slippage of about 1 % of the wind speed for undrogued SVP (Surface Velocity Program) drifters. In the context of an oil-drift study Price et al. (2006) deployed CODE (Coastal Ocean Dynamics Experiment) type drifters drogued in such a way that they were supposed to capture the upper 1 m layer velocities. Referring to a report by Niiler et al. (1997), Price et al. (2006) estimated for these drifters slip velocities of the order of 0.03 m/s. In BSHcmod+W such
velocity would match the parametrized wind drag at a wind speed of 5 m/s. Like contributions from wind drag, the estimated downwind slippage of drifters is supposedly much smaller than short term drift velocities in a tidally dominated regime but





may nevertheless have considerable impacts on drifter displacements in the long run. To strictly disentangle effects of wind drag on water masses and drifters, respectively, will hardly be possible.

Answering the second question seems again difficult. The joint analysis of drifter positions and displacements in this study gave at least some indications for possible non-ideal drifter behaviour. A period of extreme velocities far beyond what models

predict occurs for drifter #9 at the end of its journey (days 22-26). These high velocities result in a clear separation of drifter #9 from the formerly concentrated group of drifters. Probably more central for the present study is the behaviour of drifter #8. Browsing the full set of daily 25 h displacements (see SM1) suggests that there might be some problems with this drifter towards the end of its journey. From day 34 onward, drifter #8 showed a tendency to move faster than the neighbouring drifters #5 and #6 (e.g. days 34-35, day 37 or days 39-42). Strikingly, in these cases drifter #8 tended to move into directions

that are more parallel to prevailing winds. This latter observation also applies to the aforementioned behaviour of drifter #9.

About possible reasons for the deviant behaviours of drifters #8 and #9 can just be speculated. The simplest explanation would be that the different type of the two drifters (and of drifter #7, which also showed a very fast movement at the end of the time period it was tracked) distinguishes them from other drifters deployed (cf. Table 1). However, this explanation is not in accord with the fact that problems did not persist throughout the whole observational period. The fact that drifter #8 moved

jointly with drifter #6 for ten days (Fig. 5) provides strong evidence against generally different behaviours of the two drifter types.

A special behaviour of drifter #9 after about day 22 coincided with its entering a more southern region of the German Bight. For this regions Port et al. (2011) identified a higher variability of surface currents, less correlated with wind conditions, which would imply that introducing either Stokes drift or an additional wind drag could probably be a less promising approach

for model improvement. However, still the most probable explanation for the mismatch of observations and corresponding simulations is that the drifter experienced problems with its drogue. Unfortunately, drifters could not be collected at the end of their journey to check the conditions of the devices.

## 5  Conclusions

Trajectories of six surface drifters deployed in the German Bight were compared with corresponding offline simulations based

on hydrodynamic data from two independent models. While successful simulations based on archived BSHcmod currents representing a 5 m depth surface layer needed inclusion of extra wind (or wave) effects, this was not the case for simulations based on TRIM currents representing a 1 m depth surface layer. This suggests that the extensions in BSHcmod+W or BSHcmod+S, respectively, primarily acted to compensate insufficient vertical resolution in archived data. There was no convincing evidence that the drifters deployed experienced an appreciable direct wind drag.

On the other hand, it is striking that often errors in simulations based on TRIM and BSHcmod+W (or BSHcmod+S) closely resembled each other (e.g. day 8, see Figs. 7(c) and 7(g); or day 18, see Figs. 8(a) and 8(e)). This points to general problems, explanation of which probably requires analyses comprising also other aspects of hydrodynamic model output.





The present study focussed on a synoptic assessment of (mainly four) drifter trajectories overlapping in time. Expectedly, differences between synchronous drift trajectories were much larger in observations than in simulations, due to sub-grid scale processes lacking in simulations. Characteristic spatial scales in the atmosphere and the marine system differ, so that simulated fields of wind (not including sub-grid scale weather phenomena and gustiness as important drivers for drifter dispersion) and Stokes drift are even more smooth than simulated current fields. Small-scale model data misfits can therefore obviously not be remedied employing windage or Stokes drift.

Although the small number of drifters does not enable an in depth analysis, it seems that major deficiencies of simulations often manifest themselves under low or moderate wind speeds. For instance, data from days 7-9 (cf. Figs. 7(c) and 7(g) or SM3) suggest that at that time simulations underestimate currents in coastal areas. Insufficient resolution of intertidal areas could be one aspect contributing to this model deficiency. Also on days 24, 31 or 33 (cf. SM3), observed drifters moving much faster than simulated coincides with low wind conditions. All these instances correspond with changes in wind conditions and possibly also transitions between different residual current regimes (cf. Fig. 4).

On an hourly basis, contributions from windage in BSHcmod+W are often much smaller than discrepancies between simulated and observed drifter velocities (Fig. 10 or SM5), in particular under low wind conditions. When averaging over tidal cycles, relative contributions from wind forcing increase (Fig. 11). However, even small systematic errors in the simulation of oscillating tides might possibly give rise to residual currents similar in size to the contributions from windage. A finding that needs further analysis is whether nearshore residual currents underestimated in simulations indicate such inaccuracies in regions where tides increase with decreasing water depth.

Keeping in mind that we did not consider extreme events, this study did not substantiate benefits from including results of (computationally demanding) offline simulations of Stokes drift. Most of the time, directions of winds and waves coincided and the strength of Stokes drift was proportional to wind speed. Accordingly, Stokes drift could successfully be mimicked by windage, in TRIM even reasonably represented as an implicit part of parametrized momentum transfer from the atmosphere to marine currents. When winds quickly abate, increase or veer, waves adjust with a time lag, needing a certain fetch to fully develop. Although in these cases the different roles of winds and waves could be more marked, in the present study errors in atmospheric or marine circulation modelling seemed predominant. Nevertheless, fully coupled modelling of currents and waves (Staneva et al., 2016) could probably improve simulated surface currents, given that the vertical resolution is fine enough.

The incident of two drifters converging quickly and separating about ten days later provided evidence that at least in some situations an unavoidable increase in prediction uncertainty would be of the order of 3-5 km per day, regardless of however sophisticated a model used might be. Further studies would be needed to substantiate this finding in terms of its representativity and possible dependence on specific locations or atmospheric conditions. The observed separation rate happened to roughly agree with the magnitude of simulation errors we identified also for model simulations. More experiments would help identify the way to go for further model improvements.

*Data availability.* The raw data of observed drifter locations are freely available from Carrasco and Horstmann (2017).



*Author contributions.* J. Horstmann collected the field data. S. Maßmann provided BSHcmod currents and her experience regarding performance of the operational model. H. Kapitza produced the TRIM simulations. F. Schwichtenberg pre-processed the raw drifter data for comparison with model data. H. Klein produced the residual current classification based on BSHcmod surface currents. N. Groll conducted the WAM simulations. U. Callies prepared the manuscript with contributions from all co-authors.

5   *Competing interests.* The authors declare that they have no conflict of interest.

**Appendix A: Full sets of BSHcmod and TRIM simulations**

In this appendix we present simulated counterparts of all observed trajectories shown in Fig. 3. The four different model setups considered are simulations based on BSHcmod (Fig. A1), BSHcmod+W (Fig. A2), BSHcmod+S (Fig. A3) and TRIM (Fig. A4). For all figures the underlying data are provided as supplementary material.

10   *Acknowledgements.* This work was kindly supported through the Coastal Observing System for Northern and Arctic Seas (COSYNA). Wind observations at station Helgoland were provided by the Deutscher Wetterdienst (DWD). Graphs were produced using the Generic Mapping Tools software (GMT) available from www.soest.hawaii.edu/gmt/.





**Figure A1.** Simulations based on BSHcmod top layer currents, disregarding all extra effects of winds or waves. Black crosses indicate locations where simulations were started.





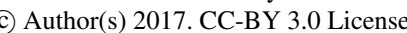

Figure A2. Simulations based on BSHcmod top layer currents plus 0.6% of 10 m wind velocity (BSHcmod+W). Black crosses indicate locations where simulations were started.





**Figure A3.** Simulations based on BSHcmod top layer currents plus 50% of surface Stokes drift (BSHcmod+S). Black crosses indicate locations where simulations were started.





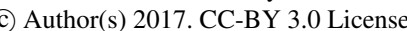

**Figure A4.** Simulations based on TRIM top layer currents, disregarding all extra effects of winds or waves. Black crosses indicate locations where simulations were started.



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



**Table 1.** Drifters deployed in May 2015

| # | Type | Start | | | End | | | Length | Dist | $\Delta$T |
|---|------|------------|-------|-------|------------|-------|-------|--------|--------|---------|
|   |      | Time (UTC) | °E | °N | Time (UTC) | °E | °N | [km] | [km] | [days] |
| 1 | MD03i | May 19 (12:31) | 7.5216 | 54.2160 | Jun 02 (21:12) | 8.8338 | 54.5180 | 1032.1 | 91.7 | 14.4 |
| (2) | MD03i | May 21 (17:13) | 7.1484 | 55.0752 | May 25 (09:47) | 7.3080 | 55.1360 | 87.4 | 12.2 | 3.7 |
| (3) | MD03i | May 21 (17:13) | 7.1480 | 55.0750 | May 25 (09:59) | 7.2526 | 55.1160 | 85.7 | 8.1 | 3.7 |
| (4) | MD03i | May 21 (17:36) | 7.1426 | 55.0786 | May 24 (15:00) | 7.2960 | 55.0626 | 66.6 | 10.0 | 2.9 |
| 5 | MD03i | May 27 (09:49) | 5.9126 | 54.3752 | Jul 15 (01:28) | 8.4680 | 55.1232 | 1264.0 | 184.4 | 48.7 |
| 6 | MD03i | May 27 (16:01) | 6.0446 | 54.2024 | Jul 20 (23:15) | 8.0944 | 55.1930 | 1467.7 | 172.1 | 54.3 |
| 7 | ODi | May 30 (08:36) | 6.7516 | 54.6712 | Jun 08 (09:59) | 8.2360 | 55.7702 | 273.2 | 154.6 | 9.1 |
| 8 | ODi | May 30 (12:09) | 6.7476 | 54.2554 | Jul 09 (19:15) | 8.5282 | 55.2812 | 1203.0 | 161.8 | 40.3 |
| 9 | ODi | May 31 (07:46) | 7.8816 | 54.0842 | Jun 24 (03:28) | 8.8360 | 54.1316 | 844.3 | 62.6 | 23.8 |

Type: Two drifter types used (cf. Fig. 1). Length: Sum of the lengths of linear segments connecting observed drifter locations. Dist: Linear distance between the first and the last drifter location observed. $\Delta$T: Days between the first and the last observation. Drifters #2, #3 and #4 travelling for only few days were ignored for this study.