# Peer review of "Surface drifters in the German Bight: Model validation considering windage and Stokes drift"

_Ocean Science, 2017_

## Referee Comment (RC1) · Anonymous Referee #1 · 22 Jun 2017

The trajectories of 6 surface drifters are compared to simulations of circulation by two numerical models in the German Bight. For one model, the numerical simulations of drifter tracks include direct downwind slip or Stokes drift estimated from a wave model. This inclusion appears necessary to compensate insufficient vertical resolution of the model. Substantial model errors, that dominate at low winds, are explained in terms of inaccurate Eulerian currents and lacking representation of the sub-grid scales processes by the models. The limit of trajectory predictability is also addressed. This paper is clear and well written, although some parts can be substantially shortened to increase readability (see below). The scientific topic is interesting and the comparison between drifter observations and simulations is done rigorously. The results show that when using a model with reduced vertical resolution, direct windage or Stokes drift

must be added in order to better predict surface drift. However, the explicit inclusion of Stokes drift does not produce an added value compared to a simple parameterization of wind-induced slip.

I recommend publication of this manuscript in Ocean Sciences after minor revision and after the authors have addressed the following specific comments.

Page 4. Paragraph 2.1. Please add the sampling frequency of the drifters. Was there a drogue presence sensor? Are you sure that the drogued-drifters kept their drogue during their entire drift? What is "R = drag area in water / drag area in air" for the drogued drifters?

Page 8. Paragraph 3.1 Line 9. Change "a view day" to " a few days"

Pages 12 to 19. Paragraph 3.2. The descriptions of the observed and simulated drifts for the different periods is too long and the reader might be bored reading all these details. I suggest to shorten these 4 pages of text by at least 50% to increase readability.

Page 20. Figure 9. Histograms represent the frequency of occurrence in selected classes of parameters. I would use the word "bar" instead of "histogram" to show distances versus time in Figure 9a and d.

---

## Referee Comment (RC2) · J. Kjellsson (Referee) · 25 Jun 2017

**1   Summary**

The manuscript describes an experiment with surface drifters and model simulations in the German Bight.  The goal is to assess the realism of the BSHmod and TRIM models.  The authors compare six observed surface drifters (tracked for about 30-40 days) with simulated drifters using the two models.  They find discrepancies between observations and models and discuss what could have caused them.

**2   Overall comments**

The manuscript is very well written with only a few misspellings and somewhat confusing sentences.  While it is good that the authors give a detailed account of the drifter experiments, the main text makes a lot of references to maps in the supplementary material, which is split into multiple files and pages.  I would recommend adding an extra plot or two in the main text where some results can be shown so that the reader does not have to go back and forth between the paper and supplementary material so often.

I recommend this paper be accepted for publication after dealing with a few minor comments.

**3   Specific comments**

**Parameters for including wind effects**
How did the authors chose the parameters in Section 2.2.3?  Did the authors try a few different values and tune the fields to match observations in this study, or was the tuning done in another study?  If the parameters were tuned in another study, please cite that study and add a comment on how well it worked. If the parameters were tuned for the data in this study, the observations and models are not really independent and a validation can not be made.  In the Discussion, the authors hint that no tuning was made. What was the motivation for choosing these values?

Page 1: Line 21: "Lagrangian transport simulations also provide . . . "
Line 22: remove "for instance"

Page 2: Line 3: "...as many of the input ..."
Line 9: "However, the Eulerian surface currents ..."
Line 10: "In cases of necessity, drifter simulations ..."
Line 12: "...5 m deep top layer. Therefore, even for an ideal ..." Line 13: "...of hy-drodynamic currents." Line 14: "...1m deep layer". Line 24-25: I found this sentence a bit confusing. "Although provided with ... for instance.". Do the authors mean that the wave model is forced with the same atm forcing as the ocean model, and the Stokes fields are added "offline", i.e. after the ocean model and wave model fields have been integrated and stored? In that case, why not write something like "Stokes drift is cal-culated from the wave model using the same wind forcing as used in the ocean model."

Page 3: Line 32-34: I found this sentence unclear. "After simulations ...25 h length." I understand drifters are split into 25 h segments, but what is meant by "different model setups explored the range of possible effects"? How were the setups different, and what were the effects?

Page 4: Line 11: "of the drifter"

Page 6: Line 8: "Eulerian model currents can usually not fully reproduce observed currents"
Line 17: How was $\alpha$ chosen? See comment above.
Line 18: remove "a" and change "parts" to "part".
Line 21: "...when model currents used do ..."
Line 22: How was $0.6\%$ chosen? See comment above.
Line 23: "(1m deep top layer)"

Line 10: I think Fig 4 is defined before Fig 3? Line 12: "A principal component analysis (PCA) was performed on the residual currents, focusing on the inner German ..."

Line 9: "a few days"
Line 12-15: I found the two sentences really hard to read. "For each ...existing observations." The time bar does not really show anything that has to do with the release of simulated drifters. I think the first sentence should be something like "The time bars show the durations of the surface drifters and different colours indicate subjectively identified drift regimes." The second sentence I think means that the time bars start at midnight, while simulations start at 13:00. Why not start the time bar at 13:00?
Line 29: "...of about 20 km from drifter 8".

Line 7: "extreme drift speeds". It is hard to judge whether the drift speeds are extreme by just looking at maps. Could the authors include a time series plot of drift velocities instead, or a probability density function for speeds? Fig. 10 and 11 show this, so I would recommend moving that plot this section and perhaps include more drifters in it.
Line 9: "moderate drift velocities". Again, hard to judge just by looking at maps.
Line 20: How were these parameter values chosen? See comment above.

Page 12:
Line 2: Somewhere here it might be good to remind the reader about how TRIM and BSHmod are different.

Line 2-4: "Although ...Eulerian currents." It is an important statement that adding

wind effect gives the correct direction for drifter 7. It would be good with a plot or at least some numbers in the text where the the drift is shown for observations, TRIM, BSHmod and BSHmod+W (only TRIM and BSHmod+W shown in Fig 7).

Line 28: "An exception is drifter 9 . . . "

Line 5: ". . . drifter 9 does not."
Line 16: "Although wind speeds can be relatively strong (not shown), strengths of 25 h . . . "
Line 20-22: "Note, however . . . locations.". I could not understand what is meant here.
Line 23: ". . . caused by the fast west-northwest movement of drifter 8, not shared by drifters 5 and 6 (SM3)."
Line 26: "A four-day period . . . "
Line 32: "A particularly fast movement of drifter 8 is observed on days 34 and 35. On day 35, drifter 8 also drifts more westward than drifter 5 and 6."

Page 19:
Line 13: "Southwesterly winds cause a transition towards a strengthened . . . "
Line 30: "Currents in TRIM representative of a surface layer of 1m depth had drift velocities similar to those observed (Fig 9)."

Page 20:
Fig 9: Why is BSHmod not shown in Fig a, or why are current speeds not shown in Fig b? It would be good to see what effect adding the wind effect actually has, i.e. what are the relative magnitudes of Eulerian currents and added wind effects? Also,

the authors should consider showing probability distributions of errors in displacement and angles in order to condense the information. Does the error in angle have a zero mean or are the errors predominant in one direction? Does one model have smaller errors than the other?

Line 9: what does "currents will generally not parallel winds" mean?
Line 15: ". . . may be one of the reasons why simulated trajectories resemble each other . . . "
Line 19-20: "Both TRIM and BSHcmod are unable to reproduce the specific . . . "
Line 24: ". . . they start with small initial separations $O(1 - 10m)$.
Line 29: "However, this separation might have been triggered . . . "
Line 34: "could imply accelerated spatial separation". Why? How? I would rather say "and relative dispersion measured using drifters of different types may not reflect the diffusivity of the flow."

Line 4: "The subsequent separation rate of about 3 km per day . . . "
Line 7: ". . . modelling was undertaken to . . . "
Line 19: Somewhere here I think a discussion is warranted about the differences in wind forcing in TRIM and BSHcmod. What is the temporal and spatial resolution of the wind data? Do they capture variations on the same spatial and temporal scales?
Line 20: Again, how were the parameters for your wind effects chosen. See comment above.

Line 1-4: Here the authors touch upon how the wind parameters were chosen, but it is not clear. Was the Stokes drift parameter chosen so that Stokes drift would be of

similar magnitudes as windage effects?

Page 25:
Line 1: "Fig 10 also shows magnitudes . . . "
Line 4-5: "Variations of maximum drift speeds indicates that movements along different branches of . . . "
Line 14: "Fig 10(a), magnitudes of drift velocities were smoothed using a 25 h moving average of hourly data"
Line 31-32: "Note that . . . wave effects." I think the authors mean they add Stokes and wind effects offline, i.e. after the Eulerian currents have been stored. Why not write "Note that the Stokes drift and windage was calculated offline and added to the Eulerian currents after the model had been integrated and the fields stored."

Line 2: What bulk formulas were used to include wind forcing in the TRIM and BSHcmod models? Same or different? Could the choice of bulk formula impact the results?
Line 9: "Two crucial and outstanding questions are a) are the drifters' behaviours representative of surface . . . "

Line 1: "To fully disentangle . . . "
Line 11: "Possible reasons for the deviant behaviours of drifters 8 and 9 can only be speculated. "

Page 28:
Line 3: Could power spectra of kinetic energy show how important the sub-grid scale

motions are?
Line 8: ". . . wind speeds in this case. "
Line 19-23: This bit is hard to understand. I think it needs some rewriting. The sentence "Keeping in mind . . . Stokes drift" should probably be split into two sentences. Also "Accordingly . . . marine currents" should probably be split as well.

---

## Referee Comment (RC3) · Anonymous Referee #3 · 4 Jul 2017

**Surface drifters in the German Bight: Model validation considering windage and Stokes drift**

This paper presents a comparison of the model BSHcmod and TRIM with six nearsurface drifters with little or no windage. The paper is well written (albeit too lengthy), the English is good and the figures are readable. I would recommend publication after minor corrections.

**Comments**

- The work shows that the BSHcmod needs the addition of either about 50% of the surface Stokes drift or about 0.6% windage. The authors concede that this is probably a reflection of the poor vertical resolution of the model as much as it reflects missing windage and/or Stokes drift. I would recommend clarifying that the Stokes drift really *is* missing whereas the windage is probably negligible as the drifter is subsurface save for the antenna.
- The TRIM results should be studied a little further. Please consider adding Stokes drift to these as well and report which percentage works best. This would help answer the question of how much the Stokes drift really should contribute to an object which sits in the upper metre or so of the water column. Ideally the Stokes drift should be vertically averaged over the upper metre (see Li et al, 2017), but a Stokes drift representative of the midpoint (say 0.5 m) will probably be close enough.
- Please mention in the text after Eq (1) that the full windage is actually a rotation (called the leeway divergence) and not simply a factor  $\beta$ .
- Section 3.2 is too lengthy. Please consider moving some of this verbiage to an appendix.

**References**

 Allen, A. and J. V. Plourde, 1999: Review of Leeway: Field Experiments and Implementation. Tech. Rep. CG-D-08-99, US Coast Guard Research and Development Center, 1082 Shennecossett Road, Groton, CT, USA, available through http://www.ntis.gov. • Li, Q., B. Fox-Kemper, Ø. Breivik, and A. Webb, 2017: Statistical Models of Global Langmuir Mixing. *Ocean Model*, **113**, 95–114, 10.1016/j.ocemod.2017.03.016.

СЗ

---

## Author Comment (AC1) · 2 Aug 2017

The comment was uploaded in the form of a supplement:
https://www.ocean-sci-discuss.net/os-2017-40/os-2017-40-AC1-supplement.pdf

———————————————

---

## Author Comment (AC2) · 2 Aug 2017

The comment was uploaded in the form of a supplement:
https://www.ocean-sci-discuss.net/os-2017-40/os-2017-40-AC2-supplement.pdf

---

## Author Comment (AC3) · 2 Aug 2017

The comment was uploaded in the form of a supplement:
https://www.ocean-sci-discuss.net/os-2017-40/os-2017-40-AC3-supplement.pdf

———————————————

---

## Author Response (AR1)

We thank J. Kjellsson and two anonymous referees most sincerely for the time and large effort they expended on a thorough review of our manuscript. Their valuable comments will definitely improve the quality of a revised paper.

In the following, the referees' comments are shown in blue.

**Response to Referee #1**

The trajectories of 6 surface drifters are compared to simulations of circulation by two numerical models in the German Bight. For one model, the numerical simulations of drifter tracks include direct downwind slip or Stokes drift estimated from a wave model. This inclusion appears necessary to compensate insufficient vertical resolution of the model. Substantial model errors, that dominate at low winds, are explained in terms of inaccurate Eulerian currents and lacking representation of the sub-grid scales processes by the models. The limit of trajectory predictability is also addressed. This paper is clear and well written, although some parts can be substantially shortened to increase readability (see below). The scientific topic is interesting and the comparison between drifter observations and simulations is done rigorously. The results show that when using a model with reduced vertical resolution, direct windage or Stokes drift must be added in order to better predict surface drift. However, the explicit inclusion of Stokes drift does not produce an added value compared to a simple parameterization of wind-induced slip.

I recommend publication of this manuscript in Ocean Sciences after minor revision and after the authors have addressed the following specific comments.

**Page 4:**

**Paragraph 2.1.**

Please add the sampling frequency of the drifters.

Positions were monitored at least every 30 minutes. Text in Section 2.1: "Three drifters could successfully be tracked for between 40 and 54 days. In order to conserve battery power, an initial sampling rate of about once every 15 min was later reduced to once every 30 min."

**Was there a drogue presence sensor?**

Drifters had no drogue pressure sensors (now mentioned in Section 2.1).

Are you sure that the drogued-drifters kept their drogue during their entire drift?

Unfortunately, we could not be sure. We address the issue in the last paragraph of the discussion. The manuscript reports indications that in particular drifters 8 and 9 may have experienced technical problems towards the end of their journey (cf. third from the last paragraph of the discussion).

**What is "R = drag area in water / drag area in air" for the drogued drifters?**

Now added in Section 2.1: "The ratio of drag area in the water to drag area outside the water was 33.2 for the MD03i and 16.9 for the ODi model, respectively."

**Page 8:**

Paragraph 3.1 Line 9. Change "a view day" to " a few days" Thanks, has been corrected.

**Pages 12 to 19.**

Paragraph 3.2. The descriptions of the observed and simulated drifts for the different periods is too long and the reader might be bored reading all these details. I suggest to shorten these 4 pages of text by at least 50% to increase readability.

We accomplished this reduction by a removal of too many details. Another option (suggested by referee #3) would have been moving parts of the material to an appendix. However, this would destroy the clear chronological order of the description, which enables a quick scan for specific events while skipping others. It would also imply a shift of corresponding figures. These figures, however, provide relevant information, which referee #2 even suggested to expand. We followed his advice and complemented Figs. 7 and 8 by a third figure (Fig. 9) so that now 12 instead of 8 example days can be shown (we compensated for that by a removal of Fig. 6, which just combined panels from different figures in the appendix). As a result, the reader is now less often forced to switch to supplementary material.

**Page 20:**

Figure 9. Histograms represent the frequency of occurrence in selected classes of parameters. I would use the word "bar" instead of "histogram" to show distances versus time in Figure 9a and d. Has been changed as suggested.

**Response to J. Kjellsson (Referee #2)**

**1** Summary**

The manuscript describes an experiment with surface drifters and model simulations in the German Bight. The goal is to assess the realism of the BSHmod and TRIM models. The authors compare six observed surface drifters (tracked for about 30-40 days) with simulated drifters using the two models. They find discrepancies between observations and models and discuss what could have caused them.

**2 Overall comments**

The manuscript is very well written with only a few misspellings and somewhat confusing sentences. While it is good that the authors give a detailed account of the drifter experiments, the main text makes a lot of references to maps in the supplementary material, which is split into multiple files and pages. I would recommend adding an extra plot or two in the main text where some results can be shown so that the reader does not have to go back and forth between the paper and supplementary material so often.

We agree that this is a problem. We tried to solve it by displaying now 12 example simulations (Figs. 7-9) rather than just eight (Figs. 7 and 8 in the original manuscript). To compensate for the additional figure, we removed the former Fig. 6, which just re-combined panels from different figures in the appendix. We believe that the additional figure much improves readability of Section 3.2, which has also been shortened by about 50 %.

I recommend this paper be accepted for publication after dealing with a few minor comments.

**3** Specific comments**

**Parameters for including wind effects**

How did the authors chose the parameters in Section 2.2.3? Did the authors try a few different values and tune the fields to match observations in this study, or was the tuning done in another study? If the parameters were tuned in another study, please cite that study and add a comment on how well it worked. If the parameters were tuned for the data in this study, the observations and models are not really independent and a validation can not be made. In the Discussion, the authors hint that no tuning was made. What was the motivation for choosing these values?

The following extra paragraph has been added at the end of Section 2.2.3, addressing this important issue:

"The assumed strengths of either wind forcing or Stokes drift resulted from trying to achieve an overall eastward displacement of simulated drifters that roughly agreed with observations. This approach must not be confused with sound model calibration, which seems impossible based on the very limited data available. Models perform differently during different periods and it is hard to distinguish, for instance, between deficiencies in the hydrodynamic model and implications of imperfect atmospheric forcing. Also independent data needed for model validation are not available. However, already the simple approach enables an appraisal of how successful drifter simulations will depend on a distinction between wind drag and Stokes drift."

At the end of the first paragraph of Section 3.2 this important aspect is emphasized once more: "A key effect of the inclusion of extra wind or wave effects is the intensification of westward transports in agreement with wind directions that occur most often. One should remember that achieving reasonable agreement between overall strengths of these transports in simulations and observations was the criterion which led to specific values we assigned to  $\alpha$  or  $\beta$  in Eq. (1) (see Section 2.2.3)."

Page 1: Line 21: "Lagrangian transport simulations also provide . . . " Changed

**Line 22: remove "for instance"**

Changed

Page 2: Line 3: ". . . as many of the input . . . " Changed Line 9: "However, the Eulerian surface currents . . . " Changed

Line 10: "In cases of necessity, drifter simulations . . . "

Changed

Line 12: "... 5 m deep top layer. Therefore, even for an ideal ... "

Changed

Line 13: ". . . of hydrodynamic currents."

Changed

Line 14: ". . . 1m deep layer".

**Changed**

Line 24-25: I found this sentence a bit confusing. "Although provided with . . . for instance.". Do the authors mean that the wave model is forced with the same atm forcing as the ocean model, and the Stokes fields are added "offline", i.e. after the ocean model and wave model fields have been integrated and stored? In that case, why not write something like "Stokes drift is calculated from the wave model using the same wind forcing as used in the ocean model."

The paragraph has been revised following also your suggestions: "Waves and resulting Stokes drift were calculated using the wind forcing also employed for hydrodynamic simulations with TRIM."

Page 3:

Line 32-34: I found this sentence unclear. "After simulations . . . 25 h length." I understand drifters are split into 25 h segments, but what is meant by "different model setups explored the range of possible effects"? How were the setups different, and what were the effects?

Has been revised: "First, full simulated trajectories are presented using currents from TRIM or BSHcmod, the latter also combined with wind drag and Stokes drift, respectively. A more detailed ..."

Page 4: Line 11: "of the drifter" Changed

Page 6:

Line 8: "Eulerian model currents can usually not fully reproduce observed currents" Sentence has been revised. Line 17: How was *α* chosen? See comment above. A new extra paragraph at the end of Section 2.2.3 now addresses this issue (see above). Line 18: remove "a" and change "parts" to "part". Changed Line 21: "... when model currents used do ... " Changed Line 22: How was 0.6% chosen? See comment above. Again, the new extra paragraph at the end of Section 2.2.3 now addresses this issue (see above). Line 23: "(1m deep top layer)" Changed

Page 7 Line 10: I think Fig 4 is defined before Fig 3? Yes, thank you for this hint! Sequence of the two figures has been changed. Line 12: "A principal component analysis (PCA) was performed on the residual currents, focusing on the inner German . . . " Changed

Page 8 Line 9: "a few days" Corrected Line 12-15: I found the two sentences really hard to read. "For each... existing observations." The time bar does not really show anything that has to do with the release of simulated drifters. I think the first sentence should be something like "The time bars show the durations of the surface drifters and different colours indicate subjectively identified drift regimes." The second sentence I think means that the time bars start at midnight, while simulations start at 13:00. Why not start the time bar at 13:00?

We agree. The modified Fig. 4 (now Fig. 3) now shows exact travel times of observed drifters. This made the rather complicated explanation you mention obsolete.

Line 29: ". . . of about 20 km from drifter 8".

Changed

**Page 11**

Line 7: "extreme drift speeds". It is hard to judge whether the drift speeds are extreme by just looking at maps. Could the authors include a time series plot of drift velocities instead, or a probability density function for speeds? Fig. 10 and 11 show this, so I would recommend moving that plot this section and perhaps include more drifters in it.

We followed this suggestion and moved former Fig. 11 to this section (now Fig. 6). The figure has also been expanded, now combining data for the four most important drifters 5, 6, 8 and 9.

Line 9: "moderate drift velocities". Again, hard to judge just by looking at maps.

A reference to Fig. 6 (former Fig. 11) is now included.

Line 20: How were these parameter values chosen? See comment above.

Described of parameter choice was improved. See our answers to the above comments.

**Page 12:**

Line 2: Somewhere here it might be good to remind the reader about how TRIM and BSHmod are different.

The following sentence was added: "It appears that combining BSHcmod currents for a 5 m depth surface layer with either windage or Stokes drift brings corresponding simulations closer to both observations (Fig. 4) and simulations based on Eulerian surface currents from TRIM with 1 m vertical resolution (Fig. A4)."

**Page 16**

Line 2-4:"Although . . . Eulerian currents." It is an important statement that adding wind effect gives the correct direction for drifter 7. It would be good with a plot or at least some numbers in the text where the the drift is shown for observations, TRIM, BSHmod and BSHmod+W (only TRIM and BSHmod+W shown in Fig 7).

The sentence was reworded to avoid misunderstanding ("*Comparing simulations based on BSHcmod+W (Fig. 8(a)) with those based on BSHcmod (SM2) reveals that the deviant simulation of drifter #7 arises from spatial variation of BSHcmod currents.*"). The deviating displacement simulated for drifter 7 is produced by differences in Eulerian currents. At the same time, adding wind forcing improves simulations of both drifter 7 and the neighboring drifters 5, 6 and 8 (as to be expected from the more large scale nature of wind fields). The additional figure for BSHcmod you are asking for is available from supplement SM2 (Please note: In SM2 we corrected all headers, now reading BSHcmod and TRIM, respectively. Former headers BSH\_TOPLAYER and TRIM\_3D\_TOPLAYER were confusing. All data remained unmodified). In addition, supplement SM4 provides the results from using Stokes drift. Unfortunately, it seems hardly feasible to include these figures for one single situation into the main manuscript.

**Page 17**

Line 28: "An exception is drifter 9 . . . "

The whole sentence was removed to shorten the manuscript as requested by referee #1.

**Page 18**

Line 5: ". . . drifter 9 does not." Sentence does no longer exist. Line 16: "Although wind speeds can be relatively strong (not shown), strengths of 25 h. . . " Again, the sentence was removed. Line 20-22: "Note, however . . . locations.". I could not understand what is meant here. Whole sentence removed. Line 23: ". . . caused by the fast west-northwest movement of drifter 8, not shared by drifters 5 and 6 (SM3)." Changed

Line 26: "A four-day period . . . "

**The phrase has been deleted.**

Line 32: "A particularly fast movement of drifter 8 is observed on days 34 and 35. On day 35, drifter 8 also drifts more westward than drifter 5 and 6."

First sentence was removed, second changed accordingly.

**Page 19:**

Line 13: "Southwesterly winds cause a transition towards a strengthened . . . "

Changed, but we kept the 'freshening', so it reads now "*Freshening southwesterly winds strengthen a cyclonic circulation.*"

Line 30: "Currents in TRIM representative of a surface layer of 1m depth had drift velocities similar to those observed (Fig 9)."

Changed into (Note that figure labels changed): "Magnitudes of TRIM surface currents, representative of a layer of 1 m depth, were generally similar to those observed (Fig. 10(a))." Page 20:

Fig 9: Why is BSHmod not shown in Fig a, or why are current speeds not shown in Fig b? It would be good to see what effect adding the wind effect actually has, i.e. what are the relative magnitudes of Eulerian currents and added wind effects? Also, the authors should consider showing probability distributions of errors in displacement and angles in order to condense the information. Does the error in angle have a zero mean or are the errors predominant in one direction? Does one model have smaller errors than the other?

We agree that the relative magnitudes of Eulerian currents and wind/wave effects are important information. Unfortunately, Fig. 9 (now Fig. 10) is already rather complex and one must be careful not to overload it. However, the manuscript provides the information you are asking for in Figs. 10 and 11 (now 12 and 6). From Fig. 10 (now 12) it can be seen that both windage and Stokes drift are much smaller than total Eulerian currents including tides. According to Fig. 11 (now 6), however, they become much more significant when considering residual currents (i.e. 25 h averages). The two figures also nicely show the mostly similar effects of windage and Stokes drift.

In response to the second part of your comment we introduced an additional figure (Fig. 11) that shows for both model approaches distributions of model errors:

**Figure 11.** Distribution of model errors in 25 h drifter simulations. Histograms are based on 164 simulations in total for drifters #5, #6, #8 and #9. Referring to drift simulations based on BSHcmod+W and TRIM, respectively, panels (a) and (b) evaluate spatial separations shown in Fig. 10(a). For the same set of 164 simulations, panels (c) and (d) evaluate directional errors from Fig. 10(d). Red lines indicate median values (4.6 km and 5.4 km in (a) and (b); -15 degrees and 7 degrees in (c) and (d)).

We doubt that the differences between the two models are statistically significant given the fact that the 164 simulations contributing to the distribution in Fig. 11 were made under very different environmental conditions. We discuss that in the revised manuscript.

Line 9: what does "currents will generally not parallel winds" mean?

Replaced by: "currents will generally not be in the direction of winds"

Line 15: "... may be one of the reasons why simulated trajectories resemble each other ... " Changed

Line 19-20: "Both TRIM and BSHcmod are unable to reproduce the specific . . . "

Changed

Line 24: ". . . they start with small initial separations O(1-10m).

The suggested change is part of a reformulation: "Ohlmann et al. (2012) start with O(5-10 m) initial separations to resolve initial non-local dispersion with exponential growth of the mean square pair separation, driven by eddies larger than the distance between the two drifters."

Line 29: "However, this separation might have been triggered . . .

**Changed**

Line 34: "could imply accelerated spatial separation". Why? How? I would rather say "and relative dispersion measured using drifters of different types may not reflect the diffusivity of the flow." The sentence was revised accordingly.

**Page 22**

Line 4: "The subsequent separation rate of about 3 km per day . . . "

Changed

Line 7: "... modelling was undertaken to ...."

Changed

Line 19: Somewhere here I think a discussion is warranted about the differences in wind forcing in TRIM and BSHcmod. What is the temporal and spatial resolution of the wind data? Do they capture variations on the same spatial and temporal scales?

Descriptions of BSHcmod and TRIM (Sections 2.2.1 and 2.2.2) have been extended accordingly. In both models atmospheric forcing is provided on an hourly basis. At this point, we added a sentence mentioning that more frequent HF radar observations might possibly enhance variability of drift simulations.

Line 20: Again, how were the parameters for your wind effects chosen. See comment above.

The parameter specification has already been addressed a couple of times (for instance, our response to your next comment below or our response to your comment at the very beginning of your list of comments). Therefore we do not see the need for getting back to this point in this paragraph that does not explicitly mention any wind parametrization.

**Page 24**

Line 1-4: Here the authors touch upon how the wind parameters were chosen, but it is not clear. Was the Stokes drift parameter chosen so that Stokes drift would be of similar magnitudes as windage effects?

The following sentences were inserted to explain this in some more detail: "The criterion we applied for selecting  $\alpha$  or  $\beta$  is that the overall eastward displacement of a drifter's location should roughly agree with that observed. A convincing confirmation of our selection was that the strength factors we chose worked consistently well for all drifters."

**Page 25:**

Line 1: "Fig 10 also shows magnitudes . . . "

Changed

Line 4-5: "Variations of maximum drift speeds indicates that movements along different branches of . . .

**Changed**

Line 14: "Fig 10(a), magnitudes of drift velocities were smoothed using a 25 h moving average of hourly data"

The important point is that the original velocity vectors were smoothed rather than just their magnitudes. After restructuring the order of figures, the sentence now occurs together with Fig. 6: *"Fig. 6 provides magnitudes of velocities for drifters #5, #6, #8 and #9, calculated from velocity vectors smoothed using a 25 h moving average of hourly data."*

Line 31-32: "Note that . . . wave effects." I think the authors mean they add Stokes and wind effects offline, i.e. after the Eulerian currents have been stored. Why not write "Note that the Stokes drift and windage was calculated offline and added to the Eulerian currents after the model had been integrated and the fields stored."

We followed this suggestion

**Page 26**

Line 2: What bulk formulas were used to include wind forcing in the TRIM and BSHcmod models? Same or different? Could the choice of bulk formula impact the results?

In Sections 2.2.1 and 2.2.2 it is now mentioned that the same parametrization of wind forcing is used in both models.

Line 9: "Two crucial and outstanding questions are a) are the drifters' behaviours representative of surface . . . "

Changed

**Page 27**

Line 1: "To fully disentangle . . . "

Changed

Line 11: "Possible reasons for the deviant behaviours of drifters 8 and 9 can only be speculated. " Changed

**Page 28:**

Line 3: Could power spectra of kinetic energy show how important the sub-grid scale motions are?

We doubt that that would be successful. One must not forget that measurements were collected under very different wind conditions so that the data would have to be partitioned accordingly. Embarking on such detailed analysis and its uncertainties would open a new discussion beyond the scope of the present paper.

Line 8: ". . . wind speeds in this case. "

**Changed**

Line 19-23: This bit is hard to understand. I think it needs some rewriting. The sentence "Keeping in mind... Stokes drift" should probably be split into two sentences. Also "Accordingly... marine currents" should probably be split as well.

The paragraph has been reformulated, sentences have been shortened.

**Response to Referee #3**

This paper presents a comparison of the model BSHcmod and TRIM with six nearsurface drifters with little or no windage. The paper is well written (albeit too lengthy), the English is good and the figures are readable. I would recommend publication after minor corrections.

We suppose that concerns about the length of the paper relate to Section 3.2. Please see our response to the last comment below.

**Comments**

• The work shows that the BSHcmod needs the addition of either about 50% of the surface Stokes drift or about 0.6% windage. The authors concede that this is probably a reflection of the poor vertical resolution of the model as much as it reflects missing windage and/or Stokes drift. I would recommend clarifying that the Stokes drift really *is* missing whereas the windage is probably negligible as the drifter is subsurface save for the antenna.

We are not sure whether we got the referee's comment right. What is meant by "Stokes drift really *is* missing"? Indeed Stokes drift wasn't taken into account when running BSHcmod and TRIM. For BSHcmod (for which a coupled setup exists) this is now explicitly mentioned at the end of the first paragraph of Sections 2.2.1: "The option to include Stokes drift from surface wave models (as described in Dick et al., 2001) is not activated operationally so that effects of Stokes drift are also not included in archived surface current data."

Another question is to which extent Stokes drift affects surface drifters in the experiment. Simulations with wave model WAM provide an estimate of the strength of Stokes drift. For more details see our response to the next comment.

• The TRIM results should be studied a little further. Please consider adding Stokes drift to these as well and report which percentage works best. This would help answer the question of how much

the Stokes drift really should contribute to an object which sits in the upper metre or so of the water column. Ideally the Stokes drift should be vertically averaged over the upper metre (see Li et al, 2017), but a Stokes drift representative of the midpoint (say 0.5 m) will probably be close enough.

Of course we also did experiments with variable strengths of windage and Stokes drift in combination with TRIM. The problem is that (as with BSHcmod) a real calibration of factors  $\alpha$  and  $\beta$  in Eq. (1) is impossible as errors are very different during different periods of time (with different wind conditions). Following recommendations of referee #2, we introduced a new paragraph at the end of Section 2.2.3 which explains the way strengths of windage and Stokes drift, respectively, were specified: "The assumed strengths of either wind forcing or Stokes drift resulted from trying to achieve an overall eastward displacement of simulated drifters that roughly agreed with observations. This approach must not be confused with sound model calibration, which seems impossible based on the very limited data available. Models perform differently during different periods and it is hard to distinguish, for instance, between deficiencies in the hydrodynamic model and implications of imperfect atmospheric forcing. Also independent data needed for model validation are not available. However, already the simple approach enables an appraisal of how successful drifter simulations will depend on a distinction between wind drag and Stokes drift."

If we would add a description of "calibration" experiments for TRIM, the same would have to be shown for BSHcmod. In our opinion, this would definitely overload the paper. And, what is more, additional effects of windage or Stokes drift for TRIM are identical with those for BSHcmod, as in both cases the same fields (calculated offline) are just superimposed upon Eulerian currents from the respective model.

Our study shows that effects of windage and Stokes drift can hardly be disentangled based on an experiment like the one we describe. Combining both effects would add another degree of freedom uncontrolled by data. Even if windage for the drifter is negligible (which seems a reasonable assumption, as the referee mentions), the poor vertical resolution of archived BSHcmod data may be remedied by kind of windage for the 1 m depth surface layer. Adding Stokes drift is an alternative option having the same effect. At the end of the first paragraph of the conclusions we added a remark regarding a similar problem when using HF radar currents: *"In a similar way, Ullman et al. (2006) attributed a bias of trajectories predicted based on HF radar currents not to a drifter leeway but rather to the fact that effective depth of HF radar measurements exceeded that of surface layer drifters."*

Although we cannot really answer the question of how much Stokes drift really contributes, it is nevertheless interesting to note that the 50% factor we chose for a reduction of surface Stokes drift is of the order of magnitude that should be expected for an object drifting in a 1 m depth surface layer. In the discussion we had already referred to a paper by Röhrs and Christensen regarding the decrease of Stokes drift with depth (last paragraph on page 23 of the original manuscript). We now added another sentence (highlighted below) stating explicitly that these values are roughly consistent with the assumed 50% factor: *"Based on these formulas, Röhrs and Christensen (2015) calculated in the context of a drifter experiment in the Barents and Norwegian Sea that an average Stokes drift of 8.9 cm/s at the surface contrasted with an average of 3.7 cm/s at 1 m depth. For the present study we neither applied theoretical profiles nor conducted an in depth model calibration.* However, in the light of the above numbers, the 50% factor  $\alpha$  in Eq. (1) we chose for BSHcmod+S seems a reasonable value for drifters representing a surface layer of about 1 m depth."

We added the reference Li et al. (2017) the referee provided. In their Eq. (23), Li et al. show Stokes drift being proportional to 10 m winds. The factor of 1.6 % is consistent with the data we provide in Fig 9(b) of the original manuscript (Fig. 10(b) in the revised manuscript).

• Please mention in the text after Eq (1) that the full windage is actually a rotation (called the leeway divergence) and not simply a factor  $\beta$ .

We added an explanation of why a drift component perpendicular to the downwind direction was not included: "Eq. (1) describes windage (or leeway) as a drag in downwind direction, neglecting any crosswind lift component. Such lift component depending on the specific overwater structure of a drifting object is crucial for search and rescue (Breivik and Allen, 2008). For surface drifters used in experiments, however, these effects should be negligible."

• Section 3.2 is too lengthy. Please consider moving some of this verbiage to an appendix.

This comment agrees with a comment of referee #1 who suggested an abbreviation of Section 3.2 by 50%. After a removal of too many details we actually achieved this. We believe that moving part of the listing to an appendix would not improve readability as it would destroy the clearly structured chronological listing, enabling a quick scan for specific events while skipping others. It would also imply a shift of corresponding figures. These figures, however, provide relevant information, which referee #2 even suggested to expand. We followed his advice and complemented Figs. 7 and 8 by a third figure (Fig. 9) so that now 12 instead of 8 example days can be shown (we compensated for that by a removal of Fig. 6, which just combined panels from different figures in the appendix). As a result, the reader is now less often forced to switch to supplementary material.

[revised manuscript text omitted]